# Isolated Causal Effects of Natural Language

Victoria Lin [1]   Louis-Philippe Morency [1]   Eli Ben-Michael [1]

## Abstract

As language technologies become widespread, it is important to understand how changes in language affect reader perceptions and behaviors. These relationships may be formalized as the *isolated causal effect* of some *focal* language-encoded intervention (e.g., factual inaccuracies) on an external outcome (e.g., readers' beliefs). In this paper, we introduce a formal estimation framework for isolated causal effects of language. We show that a core challenge of estimating isolated effects is the need to approximate all *non-focal* language outside of the intervention. Drawing on the principle of *omitted variable bias*, we provide measures for evaluating the quality of both non-focal language approximations and isolated effect estimates themselves. We find that poor approximation of non-focal language can lead to bias in the corresponding isolated effect estimates due to omission of relevant variables, and we show how to assess the sensitivity of effect estimates to such bias along the two key axes of *fidelity* and *overlap*. In experiments on semi-synthetic and real-world data, we validate the ability of our framework to correctly recover isolated effects and demonstrate the utility of our proposed measures.

## 1. Introduction

The widespread use of language technologies has given rise to an ever-expanding amount of human- and machine-generated text data. From this vast body of data emerges the opportunity to understand how information contained in language relates to real-world outcomes. Elucidating these relationships can help answer scientifically interesting questions and provide interpretability to texts and the models that generate them. For instance, what language attributes

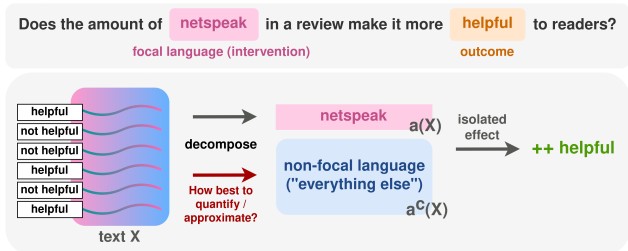

(a) The isolated causal effect of a language attribute like *netspeak* can be learned by decomposing a text $X$ into focal language $a(X)$ and non-focal language $a^c(X)$.

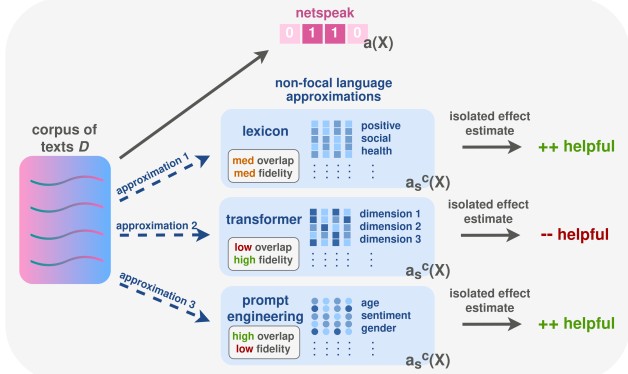

(b) Non-focal language cannot be measured directly and instead must be approximated. Due to potential *omitted variable bias*, which approximation we choose can significantly affect the isolated effect estimate.

*Figure 1.* Isolated causal effects allow us to understand how changes in language affect reader perceptions and behaviors.

(e.g., *profanity*) cause readers to perceive a passage of text as hateful? Does use of *rapport-building language* by therapists help to improve patients' mental health outcomes? Do *factual inaccuracies* propagated in machine-generated texts have negative impacts on readers' beliefs or behaviors?

The way in which language choices affect reader perceptions can be formalized as the causal effect of some language-encoded intervention—often a linguistic attribute—on an external outcome (Lin et al., 2023). However, because language is highly aliased (i.e., correlated with itself), the effect of such an intervention may be influenced by the surrounding linguistic context (Fong & Grimmer, 2023). For instance, machine-generated texts with factual inaccuracies may also contain other undesirable attributes likely to in-

[1] Carnegie Mellon University, Pittsburgh, PA, USA. Correspondence to: Victoria Lin <vlin2@andrew.cmu.edu>.

*Proceedings of the 42$^{nd}$ International Conference on Machine Learning*, Vancouver, Canada. PMLR 267, 2025. Copyright 2025 by the author(s).

fluence readers' reactions, such as inflammatory language. If the causal effect of the factual inaccuracies is estimated without accounting for aliased attributes, the resulting estimate may contain the collective effect of both the factual inaccuracies *and* some portion of the related inflammatory language—making it difficult to determine whether action should be taken to address factual inaccuracies only, inflammatory language only, or both.

Motivated by this limitation, we propose a new target of inference: the *isolated causal effect* for natural language (or *isolated effect* for brevity). We define the isolated effect as the causal effect of *only* the part of the language contained in the intervention, or the average causal effect of the focal text intervention *over all possible variations of the rest of the text* (Figure 1a).

In practice, estimating such an effect poses several major challenges. (1) We must be able to formally define and approximate not only the focal intervention but also the *non-focal language* of a text: that is, all parts of the text external to the focal intervention. (2) Incorrect modeling of the non-focal language can lead to a biased or invalid estimate of the isolated effect due to omission of key language context—a form of *omitted variable bias* (OVB) (Figure 1b). In other words, valid isolated causal effects can only be measured if the non-focal language is well-approximated. Therefore, it is important to be able to assess the robustness of the isolated effect estimate to errors or omissions in the non-focal language approximations.

To address these challenges and to provide a practical path toward estimating isolated effects, this paper introduces a formal estimation framework for isolated effects of language. Within this framework, we explore how the way we approximate non-focal language impacts isolated effect estimates due to omission of key variables, and we draw on OVB principles to define measures that assess the sensitivity of effect estimates to bias along the two key axes of *fidelity* and *overlap*.

Our experiments[1] demonstrate the validity of our framework on both semi-synthetic and real-world data. Using evaluation settings where the ground truth is known, we observe that our estimation framework is able to recover the true isolated effect across multiple interventions. We further show that our measures of overall robustness to OVB correspond closely to how well an estimator is able to recover the true effect, while fidelity and overlap provide additional insight into why an estimate is or is not correct. We suggest that these measures may be particularly useful for analysis in real-world settings where the true isolated effect is unknown.

---

[1]Our data and code are publicly available at https://github.com/torylin/isolated-text-effects.

## 2. Problem Setting

Consider a text dataset $D = \{(X_1, Y_1), \ldots, (X_n, Y_n)\}$ where texts $X_i \in \mathcal{X}$ are drawn i.i.d. from a distribution $P$, and individuals with *potential outcome* functions $Y_i(\cdot) : \mathcal{X} \to \mathbb{R}$ are drawn i.i.d. from the population $\mathcal{G}$, where $Y_i(x)$ denotes the potential outcome of individual $i$ if they were to read text $x$ (Neyman, 1923 [1990]; Rubin, 1974). We study a setting in which all confounding between the intervention and the outcome is captured in the text. Such settings are an important and common case in natural language processing (NLP) due to the widespread practice of labeling text data using external annotators who are randomly assigned to texts. This assignment mechanism in effect creates a randomized text experiment, eliminating confounding external to the text (Lin et al., 2024). Prominent NLP benchmark datasets such as SST (Socher et al., 2013), SQuAD (Rajpurkar et al., 2018), and MNLI (Williams et al., 2018), for instance, all fall under this category.

Now let $X$ be parameterized as $X = \{a(X), a^c(X)\}$, where $a(\cdot) : \mathcal{X} \to \{0, 1\}$ is the intervention—the focal language attribute for which we learn a causal effect—and $a^c(\cdot) : \mathcal{X} \to \mathbb{R}^d$ is the non-focal portion of the text. In this setting, we consider $a(\cdot)$ to be a known mapping from the text to the intervention of interest. $a(\cdot)$ is also known as a "codebook function" (Egami et al., 2022)

In naturally occurring text, $a^c(X)$ is almost certainly distributed differently when $a(X) = 1$ compared to when $a(X) = 0$. For instance, suppose $a(X)$ is *humor*, and $X$ is from a corpus of movie scripts. The non-focal (i.e., non-humor) parts of the scripts may include properties like *positive emotion* or *optimism*. As these properties are positively correlated with humor, it is much more likely when $a(X) = 1$ that they also equal 1 and when $a(X) = 0$ that they also equal 0.

To isolate the effect of *only* the focal language attribute, we must learn that effect over all possible variations of the non-focal text to be sure that no influence comes from the non-focal text. Formally, this is akin to learning the effect while enforcing the *same* non-focal text distribution $P^*$ for both conditions of $a(X)$.

**Definition 2.1** (Isolated causal effect). Let $P^*$ be some target distribution over the non-focal language. Then the isolated causal effect of $a(X)$ on $Y$ is given by:

$$\tau^* = \mathbb{E}_{Y(\cdot) \sim \mathcal{G}}[\mathbb{E}_{a^c(X)^* \sim P^*}[Y(a(X) = 1, a^c(X)^*) \\ - Y(a(X) = 0, a^c(X)^*)]]$$

We distinguish this from the *natural causal effect* defined by Lin et al. (2023), where $a^c(X)$ follows its natural distribution for both conditions of $a(X)$. The natural causal effect is the collective effect of the focal language attribute $a(X)$ *and* the parts of the non-focal language with which it

is naturally correlated.

We generalize three common assumptions for valid causal inference to the language setting:

1. *Consistency.* $Y = Y(X) = Y(a(X), a^c(X))$. The observed outcome $Y$ for an individual corresponds to the potential outcome $Y(a(X), a^c(X))$ associated with the text they actually receive.

2. *No unmeasured confounding.* $Y(x) \perp\!\!\!\perp a(X)|a^c(X)$ for all $x \in \mathcal{X}$. All confounding factors between the intervention and the outcome are captured by the non-focal portion of the text.

3. *Overlap.* $0 < P(a(X) = 1|a^c(X)) < 1$. The intervention $a(X)$ has a non-zero probability of taking either value, regardless of the non-focal portion of the text. Note that this excludes the possibility that $a(X)$ is a deterministic function of $a^c(X)$.

As we mention earlier, the assumption of no unmeasured confounding is commonly fulfilled for NLP datasets due to text-annotator assignment protocols that eliminate external confounding. In practice, it is also reasonable to believe that the overlap assumption is fulfilled since any representation of the non-focal language $a^c(X)$ is non-exhaustive, and so $a(X)$ cannot be determined solely from the representation of $a^c(X)$. The most difficult of the three assumptions to fulfill then is consistency, i.e., that observed outcomes correspond to potential outcomes. If the approximation of the non-focal language $a^c(X)$ is missing important information, then consistency may not hold. Part of the technical contribution of this paper is to characterize the implications of this assumption failing to hold.

## 3. Isolated Causal Effects of Language

In this section, we describe how to identify, estimate, and evaluate the quality of isolated effects of language. We define estimands for isolated effects, present doubly robust estimators, and discuss how approximating non-focal language during estimation can give rise to omitted variable bias. Derivations and technical results are in Appendix A.

### 3.1. Identification

The definition of the isolated causal effect requires that $a^c(X)$ follow the same target distribution $P^*$ when $a(X) = 1$ and when $a(X) = 0$, even if it does not do so naturally. To induce $a^c(X)$ to follow the same specific target distribution under both conditions of $a(X)$, we draw on *importance weighting* (IPW) principles to *transport* $a^c(X)$ from its natural distribution $P$ to the target distribution $P^*$, then supplement this with an *outcome model*.

First, let us define the transporting importance weight $\gamma$ as:

$$\gamma(a', a^c(X)) = \frac{(2a' - 1)P^*(a^c(X))}{P(a^c(X))P(a(X) = a'|a^c(X))}$$

Using the importance weight, we can identify the estimand $\tau^*$ from the observed data $D = (X, Y)$, where $X \sim P$ and $Y$ follows the resulting induced distribution on the observed responses $P_y$:

$$\begin{aligned}
\tau^* &= \mathbb{E}_D[\gamma(a(X), a^c(X))Y] \\
&= \mathbb{E}_D \left[ \frac{a(X)P^*(a^c(X))}{P(a^c(X))P(a(X) = 1|a^c(X))}Y \right] \\
&\quad - \mathbb{E}_D \left[ \frac{(1 - a(X))P^*(a^c(X))}{P(a^c(X))P(a(X) = 0|a^c(X))}Y \right]
\end{aligned}$$

This identifies the isolated effect as a difference in importance-weighted averages of the outcome between texts with and without the focal language attribute $a(X)$.

If we use only the importance weights, however, we run the risk that errors or misspecifications in the weights will lead to errors in the estimated isolated effect. Therefore, building on causal inference principles in non-language settings, we can also incorporate an outcome model $g$, defined as:

$$g(a', a^c(X)) = \mathbb{E}_{Y(\cdot) \sim \mathcal{G}}[Y(a', a^c(X))].$$

Assuming we have access to texts $X^* \sim P^*$ (or have access to the data-generating process of $P^*$), we can identify $\tau^*$ using the following *doubly robust* construction:

$$\begin{aligned}
\tau^* &= \mathbb{E}_{X^* \sim P^*}[g(1, a^c(X^*)) - g(0, a^c(X^*))] \\
&\quad + \mathbb{E}_D[\gamma(a(X), a^c(X))(Y - g(a(X), a^c(X)))] \\
&\hspace{7cm} (\tau_{DR})
\end{aligned}$$

This construction—commonly used in causal inference to identify and estimate unbiased effects and increasingly used in machine learning contexts as well—confers robustness to misspecification or mis-estimation in either the IPW term or the outcome modeling term (Robins et al., 1994; Byrd & Lipton, 2019; Kallus et al., 2022).

While $P^*$ can be any distribution over $a^c(X)$ (discussed further in Appendix A.1.3), in practice it can be unclear how to define and estimate $P^*(a^c(X))$ explicitly, as this requires characterizing a full distribution over the non-focal text probabilities. Therefore, we introduce two important realistic choices of $P^*$ that make the problem tractable.

First, we can set $P^*(a^c(X)) = P(a^c(X))$, where again $P$ is the distribution of the observed $X$. We refer to the isolated effect in this case as the *Isolated Average Treatment Effect* (IATE) in the corpus from which the texts originate.

The corresponding importance weight becomes:

$$\gamma(a', a^{\mathsf{c}}(X)) = \frac{2a' - 1}{P(a(X) = a'|a^{\mathsf{c}}(X))}.$$

Second, we can set $P^*(a^{\mathsf{c}}(X))$ to equal $P(a^{\mathsf{c}}(X)|a(X) = 1)$, the distribution of non-focal language *among the treated* (i.e., where $a(X) = 1$) in the corpus where the texts originate. We refer to this as the *Isolated Average Treatment effect on the Treated* (IATT). Estimating the IATT instead of the IATE can be beneficial in settings with potential *overlap violations*; we elaborate on this in Section 3.3. The corresponding importance weight becomes:

$$\gamma(a', a^{\mathsf{c}}(X)) = \frac{a'}{P(a(X) = 1)} \\ - \frac{(1 - a')P(a(X) = 1|a^{\mathsf{c}}(X))}{P(a(X) = 0|a^{\mathsf{c}}(X))P(a(X) = 1)}$$

### 3.2. Estimation

Having written $\tau^*$ in terms of the observable data, we describe how to estimate it in practice.

**Nuisance parameters.** To estimate $\tau_{DR}$, several nuisance parameters need to first be estimated: the outcome model $g$ and the importance weight $\gamma$. This requires approximating the non-focal language $a^{\mathsf{c}}(X)$ with a language representation. We refer to the approximation of the non-focal language using the notation $a_s^{\mathsf{c}}(X)$, where the $s$ subscript indicates that this is a mapping of the high-dimensional non-focal language $a^{\mathsf{c}}(X)$ to a lower-dimensional "short" representation space $\mathbb{R}^d$ (following the terminology used to denote a smaller feature set in Chernozhukov et al. (2024)).

With this mapping, a classifier can be trained on a separate sample to predict $a(X)$ given $a_s^{\mathsf{c}}(X)$ as input. Such a classifier outputs predicted probabilities $\widehat{P}(a(X) = a'|a_s^{\mathsf{c}}(X))$, which can be used to estimate $\widehat{\gamma}$. Using the approximation $a_s^{\mathsf{c}}(X)$, an outcome model $\widehat{g}(a(X), a_s^{\mathsf{c}}(X))$ can also be estimated on an separate sample where both texts and outcomes are available.

**Estimator.** Consider data $D = (X_i, Y_i)$, $X_i \sim P$ and $Y_i \sim P_y$; and $X_j \sim P^*$ ($i \in [n], j \in [m]$). Then the estimator for $\tau^*$ is given by

$$\widehat{\tau}_{DR} = \frac{1}{m} \sum_{j=1}^{m} [\widehat{g}(1, a_s^{\mathsf{c}}(X_j)) - \widehat{g}(0, a_s^{\mathsf{c}}(X_j))] \\ + \frac{1}{n} \sum_{i=1}^{n} \widehat{\gamma}(a(X_i), a_s^{\mathsf{c}}(X_i))(Y_i - \widehat{g}(a(X_i), a_s^{\mathsf{c}}(X_i)))$$

where the estimated $\widehat{\gamma}$ uses the appropriate probability estimates from the IATE and IATT definitions above.

Like other doubly robust estimators, $\tau_{DR}$ has a number of desirable properties. First, as long as either the weights $\gamma$

or the outcome model $g$ are correct—i.e., $\widehat{\gamma} = \gamma$ or $\widehat{g} = g$—then $\tau_{DR}$ is an unbiased estimator for $\tau^*$. Moreover, the estimator is asymptotically normal with a closed-form variance, allowing for estimation of trustworthy confidence intervals. See Kennedy (2024) for a review on these types of estimators.

### 3.3. Sensitivity to Omitted Variable Bias

**Omitted variable bias.** When representing natural language, including all "variables" in modeling is not feasible, as a full representation of language is nearly infinitely high-dimensional (e.g., a one-hot encoding of the entire English vocabulary). Instead, the non-focal language is more realistically approximated as the "short," lower-dimensional representation $a_s^{\mathsf{c}}(X)$ (e.g., a language model embedding). However, representations of language necessarily omit information relative to the original text. In this section, we link the notion of information loss from language representation to omitted variable bias. We use recent work on establishing OVB bounds for non-parametric models (Chernozhukov et al., 2024) and adapt it to a natural language setting to study the impact of omitted information in approximations of non-focal language and isolated effect estimates.

We begin by defining the *fidelity* metric $\sigma^2$ and the *overlap* metric $\nu^2$:

$$\sigma^2 = \mathbb{E}_P[(Y - g(a(X), a_s^{\mathsf{c}}(X)))^2] \\ \nu^2 = \mathbb{E}_P[\gamma(a(X), a_s^{\mathsf{c}}(X))^2]$$

where $g(a(X), a_s^{\mathsf{c}}(X))$ and $\gamma(a(X), a_s^{\mathsf{c}}(X))$ are the outcome model and importance weight that use the short non-focal language representation $a_s^{\mathsf{c}}(X)$. We call these the short outcome model and short importance weight. The fidelity metric indicates how close the short outcome model is to the true outcome model $g(a(X), a^{\mathsf{c}}(X))$, while the overlap metric indicates how well the overlap assumption for valid causal inference is fulfilled by the short importance weights. For both metrics, a smaller value is better.

Then the OVB of $\tau_{DR_s}$—that is, $\tau_{DR}$ using the short outcome model and short importance weight—is bounded:

$$|\underbrace{\tau_{DR_s} - \tau^*}_{\text{OVB}}|^2 \leq \sigma^2 \nu^2 C_Y^2 C_D^2$$

where $C_Y$ and $C_D$ are user-set sensitivity parameters for the explanatory power of omitted variables toward the outcome model and importance weight. The OVB bounds allow us to define lower and upper bounds on the isolated effect:

$$\tau_{DR}^-(C_Y, C_D), \tau_{DR}^+(C_Y, C_D) = \tau_{DR_s} \pm \sqrt{\sigma^2 \nu^2} C_Y C_D$$

**The fidelity-overlap tradeoff.** A tradeoff between fidelity and overlap emerges when choosing how to approximate

the non-focal language $a^c(X)$ as $a_s^c(X)$. If $a_s^c(X)$ is a high-dimensional dense representation like a language model embedding, then model fidelity is likely to be good, as the short outcome model $g(a(X), a_s^c(X))$ has plenty of information with which to make predictions. However, representations with good fidelity are also more prone to overlap violations. While we assume in Section 2 that strict overlap is fulfilled, $P(a(X) = a'|a_s^c(X))$ that are very close to 0 and 1 ("near overlap violations") can still skew the importance weights $\gamma$ to extreme values, heavily impacting effect estimates. These near overlap violations occur more often if $P(a(X) = a'|a_s^c(X))$ is computed using high-dimensional dense representations for $a_s^c(X)$, as the greater number of dimensions makes it more likely that certain values of $a_s^c(X)$ are almost exclusively seen with either $a(X) = 1$ or $a(X) = 0$.[2]

Importantly, the fidelity-overlap tradeoff can be balanced by considering the overall *robustness value* of the isolated effect that uses the non-focal language representation $a_s^c(X)$:

$$RV = \left| \frac{\tau_{DR_s}}{\sigma\nu} \right|$$

Intuitively, the robustness value is a measure of the effect estimate's trustworthiness: it indicates how robust the estimate is to OVB. The robustness value can be seen as the amount of explanatory power that can be lost from approximating $a^c(X)$ as $a_s^c(X)$ before the isolated effect is no longer correct in sign (positive or negative). A larger robustness value corresponds a higher tolerance to OVB.

We estimate $\hat{\sigma}^2$, $\hat{\nu}^2$, and the robustness value from the data using debiased estimators (Appendix A.3). We note that under this type of estimation, it is possible for the estimated $\hat{\nu}^2$ to be negative; this indicates that something may have gone wrong with importance weight estimation (potentially a severe overlap violation).

Finally, we emphasize that while OVB may correspond to how close an effect estimate is to the ground truth, it is a complementary measure. By assessing effect estimates through the lens of each metric—fidelity, overlap, and robustness value—we gain greater insight into how and why different non-focal language approximations can influence isolated effect estimation.

## 4. Experiments

To assess the validity of isolated effects estimated using our framework, we examine how well we can recover the true isolated effect $\tau^*$ with our estimator $\hat{\tau}_{DR}$. We evaluate on two natural language datasets—one semi-synthetic and one real-world—in which true isolated effects are known.

---

[2]The IATT is less susceptible to overlap violations than the IATE, as the IATT requires only that $P(a(X) = 1|a_s^c(X)) < 1$.

### 4.1. Datasets

**Amazon (*partially controlled setting*).** The Amazon dataset (McAuley & Leskovec, 2013) consists of reviews from the Amazon e-commerce site, each with a number of "helpful" votes. To reduce unmeasured factors in the data, we generate a new semi-synthetic outcome $Y$ by predicting the number of helpful votes as a linear function of $a(X), a_s^c(X)$, then adding noise. Here, $a(X), a_s^c(X)$ encode the 10 categories from the lexicon LIWC (see Section 4.2.1) that are most predictive of vote count. These categories are binarized to take the value 1 if the category is present in the text and 0 otherwise. We note that while the semi-synthetic construction allows us to control the outcome $Y$, we do not have influence over the joint distribution $P(a(X), a_s^c(X))$.

In this partially controlled data setting, we know both (1) the true isolated effect of each of the 10 lexical categories and (2) that the outcome model $g$ can be fully learned from the text. Combined, these allow us to evaluate whether our estimator $\hat{\tau}_{DR}$ is able to recover the true isolated effect under best-case conditions. To allow for controlled evaluation under more challenging conditions, we also generate a second semi-synthetic $Y$ where helpful votes are predicted as a *non-linear* function of $a(X), a_s^c(X)$; this setting is discussed in Section B.2.

**Semaglutide vs. Tirzepatide (SvT) (*real-world setting*).** Here, we consider a slightly different setting in which the intervention and outcome are both encoded in text (in contrast to the setting where the intervention is encoded in the text and the outcome is a numerical value external to the text). The SvT dataset (Dhawan et al., 2024) consists of posts from weight-loss communities on the social media site Reddit that mention one of two weight-loss medications: semaglutide or tirzepatide. From these posts, Dhawan et al. extracted the language-encoded binary intervention $a(X)$ (which weight-loss medication the user took) and binary outcome $Y$ (whether the user lost more than 5 percent of their starting body weight). The dataset further includes a "ground truth" causal effect from a clinical trial on the effects of semaglutide versus tirzepatide at various doses (Frías et al., 2021). Dhawan et al. used weight loss under 5 mg tirzepatide versus 1 mg semaglutide as the true effect. We compute confidence intervals for this true effect using information available in the clinical trial.

This dataset allows us both to evaluate the validity of isolated effect estimates in a realistic setting and to assess how different approximations of $a^c(X)$ can impact our estimates.

### 4.2. Implementation

#### 4.2.1. APPROXIMATING NON-FOCAL LANGUAGE

To construct a non-focal language approximation $a_s^c(X)$, we explore a number of language representations varying in

complexity. In this section, we describe each representation and discuss how it might fare in the fidelity-overlap tradeoff. Implementation details can be found in Appendix C.2.

**Lexicon.** One simple language representation is a vector of interpretable categories encoded by a *lexicon*, which maps words in its vocabulary to those categories. These categories are usually relatively few in number, so the dimensionality of the category vector is fairly low. However, lexicons are limited by their vocabulary and are also unable to capture context or sentence-level meaning. Therefore, we expect that lexicon-derived non-focal language representations may have good overlap but poor model fidelity. We use two well-known lexicons in our experiments: the human expert-designed LIWC (Pennebaker et al., 2015) and the semi-automatically generated Empath (Fast et al., 2016).

**Language model embedding.** Sentence embeddings from transformer-based language models are among the most commonly used language representations for machine learning. These embeddings are information-rich in content and syntax and perform excellently on a wide variety of tasks. However, language model embeddings tend to be relatively high-dimensional compared to lexicons, and as a result, embedding-derived non-focal representations may achieve good model fidelity but suffer from overlap violations. In our experiments, we use embeddings extracted from the pre-trained transformers BERT (Devlin et al., 2019), RoBERTa (Liu et al., 2019), MPNet (Song et al., 2020), and MiniLM (Wang et al., 2020). For RoBERTa and MPNet, we also use singular value decomposition (SVD) to create a lower-dimensional version of each embedding. We refer to these smaller 200-dimension representations as RoBERTa+SVD and MPNet+SVD.

**SenteCon.** SenteCon is a language representation in which a lexicon-based layer is constructed over language model embeddings (Lin & Morency, 2023). Like lexicon representations, a SenteCon representation consists of a vector of interpretable categories where each category is associated with a numerical weight. Because the categories are derived from an existing lexicon, the dimensionality of the Sente-Con category vector should also be low. SenteCon does not rely exclusively on a pre-defined vocabulary and is able to capture sentence context. Consequently, we expect that a SenteCon-derived non-focal language representation will have reasonable model fidelity while also not being significantly affected by overlap violations. In our experiments, we use two different base lexicons for SenteCon (LIWC or Empath). We refer to these variants as SenteCon-LIWC and SenteCon-Empath.

**LLM prompting.** As large language model (LLM) capabilities continue to expand, it has become possible to extract attributes from a text passage simply by prompting an LLM. For instance, we might ask an LLM to tell us, based on

the information contained in a paragraph, the age of the writer or if they have any health conditions. Using this form of prompting on GPT-3.5, Dhawan et al. (2024) extract a set of 10 health-related attributes from Reddit posts in the SvT dataset. Of these, we exclude 3 attributes from which weight loss can be directly computed. We treat the remaining 7 discrete variables as a type of language representation and therefore an approximation of the non-focal language.

### 4.2.2. MODELING AND ESTIMATION

For each non-focal language representation $a_s^c(X)$, we use 5-fold cross-fitting to train an outcome model $\widehat{g}$ to predict $Y$ given $a_s^c(X)$ and a classifier to predict $a(X)$ given $a_s^c(X)$. We use $\widehat{P}(a(X) = a'|a_s^c(X))$ from this classifier to estimate $\widehat{\gamma}$. Within the training folds, we conduct 5-fold cross-validation to select model hyperparameters. For the linear-outcome case of the Amazon dataset, we use a logistic regression classifier and a linear regression outcome model (and neural networks for the nonlinear case). For the SvT dataset, we use gradient boosting models for both our classifier and outcome model. Additional model details, including libraries and hyperparameters, are available in Appendix C.3.

With $\widehat{g}$ and $\widehat{\gamma}$ estimated, we are able to compute $\widehat{\tau}_{DR}$ on the estimation folds, which we call $\mathcal{D}_{\text{estimate}}$. When estimating the IATE, we directly use $\mathcal{D}_{\text{estimate}}$ as the source of texts $X^* \sim P^*$ for the outcome modeling term, as the target distribution is equal to the observed data distribution. When estimating the IATT, we draw $X^*$ from the subset of $\mathcal{D}_{\text{estimate}}$ where $a(X) = 1$. We estimate the IATE for the Amazon dataset and the IATT for the SvT dataset to maintain better overlap.

We compare our isolated effect estimates against a *naive estimator* that does not isolate the focal attribute from the non-focal portion of the text (i.e., an estimator of the natural effect). In general, we expect this not to correctly recover the true isolated effect since it has no adjustment for isolation.

## 5. Results and Discussion

In this section, we evaluate how well our method is able to recover true isolated causal effects in the Amazon and SvT datasets. Following this evaluation, we more closely examine the relationship between isolated effect estimation and omitted variable bias.

### 5.1. Amazon Dataset

In the Amazon dataset, we examine the isolated effects of the 10 predictive LIWC categories used to construct the semi-synthetic outcome. This section discusses the linear-function outcome, but the same trends appear in the nonlinear case (results in Section B.2).

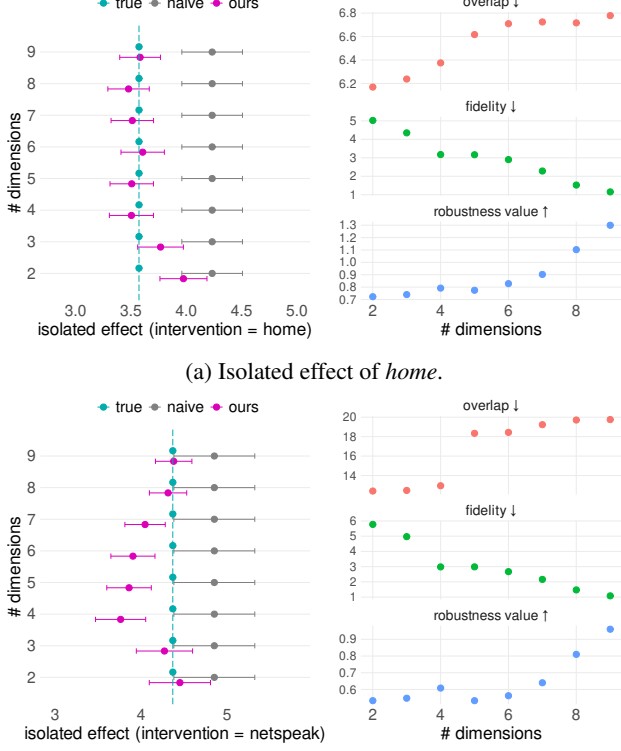

(a) Isolated effect of *home*.

(b) Isolated effect of *netspeak*.

*Figure 2.* Isolated causal effects of linguistic attributes on helpfulness in the Amazon dataset. Error bars correspond to 95% confidence intervals.

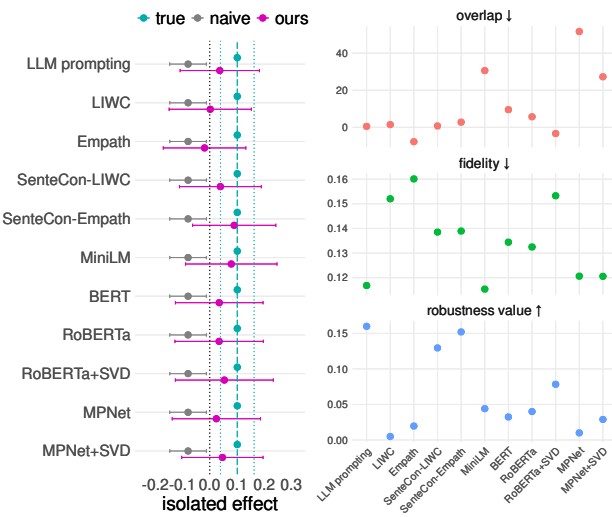

*Figure 3.* Isolated causal effect of weight-loss medication in the SvT dataset. Error bars denote 95% confidence intervals. The blue dotted lines surrounding the true effect mark its 95% confidence interval. Representations $a_s^c(X)$ are ordered loosely by complexity, with less complex representations appearing closer to the top

.

Across the 10 categories, we iteratively set each category to be $a(X)$ and estimate its isolated effect while using the remaining lexical categories as $a^c(X)$. To explore the fidelity-overlap tradeoff under controlled conditions, we evaluate our isolated effect estimate and our three OVB-derived metrics—$\widehat{\sigma}^2$, $\widehat{\nu}^2$, and robustness value—under different choices of $a_s^c(X)$. For each intervention, we restrict the number of remaining lexical categories used as $a_s^c(X)$, beginning with 2 categories and ending with 9.

Over multiple interventions (Figures 2a, 2b; additional results in Appendix B.1), we observe that as the dimensionality (number of categories) of $a_s^c(X)$ increases, so does the proximity of the isolated effect estimate to the ground truth. Moreover, the behavior of the fidelity and overlap metrics $\widehat{\sigma}^2$ and $\widehat{\nu}^2$ is also consistent with expectations. As dimensionality increases, $\widehat{\sigma}^2$ decreases, indicating that outcome model fidelity is improving. At the same time, $\widehat{\nu}^2$ increases, consistent with worsening overlap.

We further see that robustness values increase with $a_s^c(X)$ dimensionality, suggesting that gains in model fidelity outweigh losses in overlap. This may not be the case for all datasets. As this dataset does not experience significant problems with overlap (as seen from the limited range of

$\widehat{\nu}^2$, particularly in Figure 2a), it seems that outcome model performance gains are more significant in this case.

### 5.2. Semaglutide vs. Tirzepatide Dataset

In the SvT dataset, we use the intervention and outcome from Dhawan et al. (2024). We treat the Reddit post text as the non-focal language $a^c(X)$, and we explore how each of the non-focal language representations in Section 4.2.1 impacts effect estimation.

We first observe that all of our isolated effect estimates have wide 95% confidence intervals that include both the true isolated effect and 0 (Figure 3). Looking solely at the point estimates, we see that almost all of the representations yield positive isolated effect estimates that are consistent with the ground truth. Of these, SenteCon-Empath comes closest to recovering the true isolated effect, with MiniLM a close second—but a large amount of uncertainty remains. As a result, we may not be able to use the point estimates alone to determine which representation best approximates non-focal language.

We look instead to fidelity and overlap, where we observe interesting behavior. The high-dimensional MPNet embedding has a much larger $\widehat{\nu}^2$ than any other representation, suggesting a near overlap violation. Interestingly, BERT and RoBERTa—which have the same dimensionality as MPNet—exhibit much better overlap than MPNet, possibly due to the additional optimization of MPNet for sentence-level tasks in the library used to extract its embedding. We

also see that several representations, such as the lexicon Empath and the dimensionality-reduced RoBERTa+SVD, have negative $\widehat{\nu}^2$s. Because doubly robust estimators like the one we use for $\widehat{\nu}^2$ do not necessarily satisfy criteria like being non-negative in noisy settings, we hypothesize that these negative values (which are small in magnitude) may be due to noise in estimation. Fidelity, on the other hand, is much more consistent across all representations. In general, fidelity is expected to improve (i.e., $\sigma^2$ should decrease) with the dimensionality of the representation, as higher-dimensional representations are likely to contain more information; however, this may be negated by strong regularization in the outcome model. These results suggest that the fidelity-overlap tradeoff depends on some notion of representation complexity that may go beyond the dimensionality of the representation alone.

Finally, we find that the two representations with the highest robustness values are LLM prompting, which produces a positive but conservative effect estimate, and SenteCon-Empath, which produces an estimate very close to the true effect. MiniLM, which like SenteCon-Empath has a point estimate close to the clinical benchmark, has only a middling robustness value due to poor overlap. The robustness of LLM prompting is not unexpected: Dhawan et al. (2024) carefully designed their prompting procedure to extract discrete variables for the specific task of estimating the effect of tirzepatide versus semaglutide on weight loss, so we expect this representation to yield good results. Importantly, however, the SenteCon-Empath representation—which is *not* specifically designed for this task—has a similarly high robustness value, suggesting that equally effective representations can be found without requiring extensive human design effort.

Our results also illustrate the potential of dimensionality reduction methods like SVD in non-focal language approximation. Both RoBERTa and MPNet benefit from singular value thresholding across all OVB metrics: overlap improves, fidelity remains similar, and robustness value increases. Moreover, after SVD is applied, both representations' effect point estimates move from *outside* the true effect confidence interval to *inside* the true effect confidence interval. These results suggest that dimensionality reduction can significantly improve the utility of high-dimensional non-focal language representations. Given the low computational and human overhead of these unsupervised post-processing techniques, they may often be worth trying.

**A closer look at OVB and robustness.** While robustness values can be compared among non-focal language representations—providing some sense of how relatively robust each corresponding effect estimate is to bias—it is not immediately clear whether even the representation with the best robustness value is robust in absolute terms.

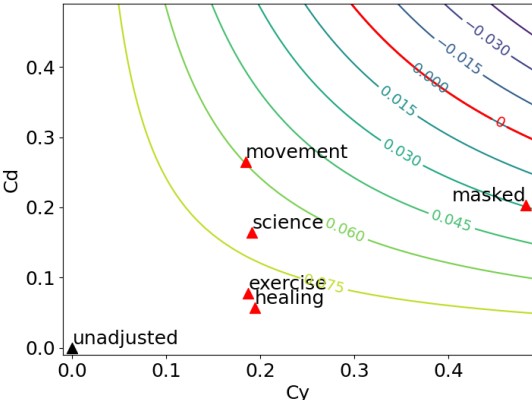

*Figure 4.* OVB lower bound of SenteCon-Empath isolated effect estimate in the SvT dataset. "Unadjusted" marks the point estimate lower bound without OVB.

One way of understanding the scale of a robustness value is to calibrate it using the explanatory power lost when intentionally omitting variables known to have an influence on the effect estimate. We focus on one representation—SenteCon-Empath in the SvT dataset—and the lower OVB bound of its associated isolated effect estimate. We recall that this bound corresponds to the least possible value of the effect point estimate at a given level of OVB.

In Figure 4, we plot this bound against the sensitivity parameters $C_Y$ and $C_D$, which denote the explanatory power of omitted variables toward the outcome model $g$ and the importance weight $\gamma$, respectively. This contour plot shows how the lower OVB bound of the effect estimate changes as the hypothetical explanatory power of the variables omitted from the SenteCon-Empath representation increases. We color the 0 contour red to highlight the significance of the lower OVB bound "crossing" from positive to negative as $C_Y$ and $C_D$ increase. Once the bound is negative, we can no longer be certain that the point estimate of our isolated effect is positive.

We then plot four red triangles to mark the explanatory power lost by explicitly omitting the category with that name (*movement*, *science*, *exercise*, or *healing*) from the SenteCon-Empath representation.[3] We choose categories we believe to be relevant to the intervention and outcome. For each category omitted, we see the loss of explanatory power brings the point estimate closer to the 0 contour but is not nearly enough to cross it. Turning then to the *masked* marker, we look at the explanatory power lost by omitting key information from the non-focal text $a^c(X)$ itself. We create a version of each Reddit post where we mask medication type, body weight, and terms like "gain" and "loss."

---

[3]These values can be computed explicitly by re-fitting the outcome models and importance weights (Appendix C.4).

The resulting SenteCon-Empath representations experience a much larger drop in explanatory power, but the lower bound of the estimate still remains positive. These results suggest that the isolated effect estimate is robust (though noisy, as the wide confidence intervals may indicate), as the lower bound on the point estimate remains positive even under levels of OVB comparable to removing relevant lexical categories or masking key information from the text.

## 6. Related Work

**Causal effects of text.** Our work on isolated causal effects is situated within a recent literature on estimating text-based causal effects. Egami et al. (2022) describe a conceptual *codebook* framework for causal inference using text. Building on this, Fong & Grimmer (2023) conduct randomized text experiments where texts are programmatically generated from pre-specified attributes. Lin et al. (2023) further propose a method for transporting natural (i.e., non-isolated) causal effects from randomized text experiments to potentially non-randomized target distributions.

Most directly related to our work are several methods for estimating isolated effects of natural language using specific language representation pipelines. Though these works do not explicitly distinguish isolated and natural effects, their estimands are defined such that they are isolated, and so we view these methods as complementary to ours. Pryzant et al. (2021) estimate the effect of a proxy linguistic attribute from observational data, where all other language information is represented by an embedding from a transformer trained to capture confounding. Dhawan et al. (2024) estimate effects from observational data using the LLM prompting approach we describe earlier. Finally, a recent paper by Imai & Nakamura (2024) estimates text effects via a randomized experiment in which LLM-generated texts are shown to human respondents; text representations can then be extracted directly from the generating LLM.

**Omitted variable bias.** The diversity of language representations that can be used in text-based causal inference highlights the strong need for a way to understand the quality of representations and effect estimates. Our OVB-based metrics provide a way to do this flexibly across any data setting, which is important for real-world data where the ground truth is not known. Our metrics draw directly on the Chernozhukov et al. (2024) operationalization of OVB, which in turn builds on a history of foundational work on OVB (Goldberger, 1991; Frank, 2000; Angrist & Pischke, 2009; Oster, 2019; Cinelli & Hazlett, 2019).

Noting the importance of covariate representation in causal inference, Clivio et al. (2024b) have proposed learning representations that minimize information loss when balancing covariates, which can help to improve overlap (Clivio et al.,

2024a). While these methods are not developed specifically for text, the parallels between general covariate representation and language representation are evident.

## 7. Conclusion

In this paper, we propose a framework for estimating the *isolated causal effect* of a *focal* language-based intervention. Estimating isolated effects is challenging because it requires us to model not only the focal intervention but also the *non-focal* language of the text. We introduce measures for assessing the sensitivity of isolated effect estimates to omitted variable bias in their non-focal language approximations along the axes of *fidelity* and *overlap*. We demonstrate the ability of our framework to correctly recover isolated effects across multiple language-encoded interventions, and we explore how the way we approximate non-focal language impacts fidelity, overlap, and the robustness of the effect estimates themselves.

Our results point to several avenues for future research. This paper studies a setting in which confounding is contained fully in the text. Though NLP datasets are not often released with external confounding data like information about annotators, it may still be interesting to study the case where external confounding is present and measured. Additionally, in this paper we treat the focal language-encoding function $a(\cdot)$ as an accurate parameterization of the intervention of interest, but if $a(\cdot)$ does need to be estimated, then estimation error can lead to additional bias. Characterizing and counteracting this is important future work. Finally, our findings on OVB and robustness suggest a compelling line of research on learning representations—perhaps not only of language—that optimize the fidelity-overlap tradeoff to minimize omitted variable bias, making them explicitly suitable for the task of causal inference.

## Impact Statement

**Broader impact.** Recent advances in NLP have dramatically increased the availability of language data and models for common users. The resulting proliferation of texts and models has raised potential ethical concerns around factual inaccuracies (Monteith et al., 2024; Zhou et al., 2023), bias (Wan et al., 2023; Ferrara, 2024), and the black-box internals of models (Guidotti et al., 2018; McDermid et al., 2021). These concerns emphasize the growing need to understand the impacts of texts and language models on the readers that consume them. Our work on isolated causal effects builds toward this goal.

**Ethical considerations.** The empirical analysis contained in this work relies partially on representations from pre-trained large language models, which may encode biases from their training data. Interpretations of causal effects that

rely on such representations should consider these biases. We additionally acknowledge the environmental impact of training the language models used in this work.

## Acknowledgements

This material is based upon work partially supported by the National Institutes of Health (awards R01MH125740, R01MH132225, R21MH130767, and U01MH136535). Victoria Lin is supported by a Meta Research PhD Fellowship. Any opinions, findings, conclusions, or recommendations expressed in this material are those of the author(s) and do not necessarily reflect the views of the sponsors, and no official endorsement should be inferred.

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

# A. Derivations and Proofs

## A.1. ATE and ATT

### A.1.1. IDENTIFYING THE ESTIMAND

*Proof.* Equivalence of $\tau_{DR}$ and $\tau^*$.

We have $X \sim P$, $Y \sim P_y$, $D = (X, Y)$, and $X^*$ where $a^{\mathsf{c}}(X^*) \sim P^*$. Because we set the value of $a(X^*)$, we use the notation $X^* \sim P^*$ and $a^{\mathsf{c}}(X^*) \sim P^*$ interchangeably. In principle, text comes from a finite sample space, so we use summations and probability mass functions to describe it.

We want to show the following:

$$
\begin{aligned}
\tau_{DR} &= \mathbb{E}_D\left[\gamma(a(X), a^{\mathsf{c}}(X))(Y - g(a(X), a^{\mathsf{c}}(X)))\right] + E_{X^* \sim P^*}[g(1, a^{\mathsf{c}}(X^*)) - g(0, a^{\mathsf{c}}(X^*))] \\
&= \mathbb{E}_{Y(\cdot) \sim \mathcal{G}}\left[\mathbb{E}_{a^{\mathsf{c}}(X^*) \sim P^*}\left[Y(a(X) = 1, a^{\mathsf{c}}(X^*)) - Y(a(X) = 0, a^{\mathsf{c}}(X^*))\right]\right] \\
&= \tau^*
\end{aligned}
$$

First, notice

$$
\begin{aligned}
\gamma(a', a^{\mathsf{c}}(X)) &= \frac{(2a' - 1)P^*(a^{\mathsf{c}}(X))}{P(a(X) = a'|a^{\mathsf{c}}(X))P(a^{\mathsf{c}}(X))} \\
&= \frac{(2a' - 1)P^*(a^{\mathsf{c}}(X))P(a(X) = a')}{P(a(X) = a'|a^{\mathsf{c}}(X))P(a^{\mathsf{c}}(X))P(a(X) = a')} \\
&= \frac{(2a' - 1)P^*(a^{\mathsf{c}}(X))}{P(a^{\mathsf{c}}(X)|a(X) = a')P(a(X) = a')} \\
&= \frac{(2a' - 1)P^*(a^{\mathsf{c}}(X))}{P(a^{\mathsf{c}}(X), a(X) = a')} \\
&= \begin{cases} \frac{P^*(a^{\mathsf{c}}(X))}{P(a^{\mathsf{c}}(X), a(X)=1)} & \text{if } a' = 1 \\ -\frac{P^*(a^{\mathsf{c}}(X))}{P(a^{\mathsf{c}}(X), a(X)=0)} & \text{if } a' = 0 \end{cases}
\end{aligned}
$$

Now consider the first term of $\tau_{DR}$:

$$
\begin{aligned}
&\mathbb{E}_D\left[\gamma(a(X), a^{\mathsf{c}}(X))(Y - g(a(X), a^{\mathsf{c}}(X)))\right] = \mathbb{E}_{Y(\cdot) \sim \mathcal{G}}\left[\mathbb{E}_X\left[\gamma(a(X), a^{\mathsf{c}}(X))(Y - g(a(X), a^{\mathsf{c}}(X)))\right]\right] \\
&= \mathbb{E}_{Y(\cdot) \sim \mathcal{G}}\left[\mathbb{E}_X\left[\frac{P^*(a^{\mathsf{c}}(X))}{P(a^{\mathsf{c}}(X), a(X) = 1)}(Y - g(1, a^{\mathsf{c}}(X)))\mathbb{1}\{a(X) = 1\}\right.\right. \\
&\qquad\qquad\left.\left. -\frac{P^*(a^{\mathsf{c}}(X))}{P(a^{\mathsf{c}}(X), a(X) = 0)}(Y - g(0, a^{\mathsf{c}}(X)))\mathbb{1}\{a(X) = 0\}\right]\right] \\
&= \mathbb{E}_{Y(\cdot) \sim \mathcal{G}}\left[\mathbb{E}_X\left[\frac{P^*(a^{\mathsf{c}}(X))}{P(a^{\mathsf{c}}(X), a(X) = 1)}(Y - g(1, a^{\mathsf{c}}(X)))\mathbb{1}\{a(X) = 1\}\right]\right] \\
&\qquad - \mathbb{E}_{Y(\cdot) \sim \mathcal{G}}\left[\mathbb{E}_X\left[\frac{P^*(a^{\mathsf{c}}(X))}{P(a^{\mathsf{c}}(X), a(X) = 0)}(Y - g(0, a^{\mathsf{c}}(X)))\mathbb{1}\{a(X) = 0\}\right]\right]
\end{aligned}
$$

Now we can rewrite

$$\mathbb{E}_{Y(\cdot)\sim\mathcal{G}}\left[\mathbb{E}_X\left[\frac{P^*(a^{\mathrm{c}}(X))}{P(a^{\mathrm{c}}(X),a(X)=a')}(Y-g(a',a^{\mathrm{c}}(X)))\mathbb{1}\{a(X)=a'\}\right]\right]$$

$$=\mathbb{E}_{Y(\cdot)\sim\mathcal{G}}\left[\mathbb{E}_X\left[\sum_{x\in\mathcal{X}}\frac{P^*(a^{\mathrm{c}}(x))}{P(a^{\mathrm{c}}(x),a(x)=a')}(Y(a',a^{\mathrm{c}}(x))\right.\right.$$

$$\left.\left.-g(a',a^{\mathrm{c}}(x)))\mathbb{1}\{a(x)=a'\}\mathbb{1}\{a^{\mathrm{c}}(X)=a^{\mathrm{c}}(x),a(X)=a(x)\}\right]\right]$$

$$=\mathbb{E}_{Y(\cdot)\sim\mathcal{G}}\left[\sum_{x\in\mathcal{X}}\frac{P^*(a^{\mathrm{c}}(x))}{P(a^{\mathrm{c}}(x),a(x)=a')}(Y(a',a^{\mathrm{c}}(x))\right.$$

$$\left.-g(a',a^{\mathrm{c}}(x)))\mathbb{E}_X\left[\mathbb{1}\{a(x)=a'\}\mathbb{1}\{a^{\mathrm{c}}(X)=a^{\mathrm{c}}(x),a(X)=a(x)\}\right]\right]$$

$$=\mathbb{E}_{Y(\cdot)\sim\mathcal{G}}\left[\sum_{x\in\mathcal{X}}\frac{P^*(a^{\mathrm{c}}(x))}{P(a^{\mathrm{c}}(x),a(x)=a')}(Y(a',a^{\mathrm{c}}(x))-g(a',a^{\mathrm{c}}(x)))P(a^{\mathrm{c}}(x),a(x)=a')\right]$$

$$=\mathbb{E}_{Y(\cdot)\sim\mathcal{G}}\left[\sum_{x\in\mathcal{X}}P^*(a^{\mathrm{c}}(x))(Y(a',a^{\mathrm{c}}(x))-g(a',a^{\mathrm{c}}(x)))\right]$$

$$=\mathbb{E}_{Y(\cdot)\sim\mathcal{G}}\left[\mathbb{E}_{X^*}\left[Y(a',a^{\mathrm{c}}(X^*))-g(a',a^{\mathrm{c}}(X^*))\right]\right]$$

$$=E_{X^*}\left[\mathbb{E}_{Y(\cdot)\sim\mathcal{G}}\left[Y(a',a^{\mathrm{c}}(X^*))-\mathbb{E}_{Y(\cdot)\sim\mathcal{G}}[Y(a',a^{\mathrm{c}}(X^*))]\right]\right]$$

$$=0$$

Then consider the second term of $\tau_{DR}$:

$$E_{X^*\sim P^*}[g(1,a^{\mathrm{c}}(X^*))-g(0,a^{\mathrm{c}}(X^*))]=E_{X^*\sim P^*}[\mathbb{E}_{Y(\cdot)\sim\mathcal{G}}[Y(1,a^{\mathrm{c}}(X^*))]-\mathbb{E}_{Y(\cdot)\sim\mathcal{G}}[Y(0,a^{\mathrm{c}}(X^*))]]$$
$$=E_{a^{\mathrm{c}}(X^*)\sim P^*}[\mathbb{E}_{Y(\cdot)\sim\mathcal{G}}[Y(1,a^{\mathrm{c}}(X^*))-Y(0,a^{\mathrm{c}}(X^*))]]$$

Then putting everything together,

$$\tau_{DR}=0-0+E_{Y(\cdot)\sim\mathcal{G}}[E_{a^{\mathrm{c}}(X^*)\sim P^*}[Y(1,a^{\mathrm{c}}(X^*))-Y(0,a^{\mathrm{c}}(X^*))]]=\tau^*$$

$\square$

### A.1.2. $\gamma$ FOR TWO SPECIAL CASES

(1) IATE: With $P^*(a^{\mathrm{c}}(X))=P(a^{\mathrm{c}}(X))$, we can rewrite $\gamma$:

$$\gamma(a',a^{\mathrm{c}}(X))=\frac{(2a'-1)P^*(a^{\mathrm{c}}(X))}{P(a^{\mathrm{c}}(X))P(a(X)=a'|a^{\mathrm{c}}(X))}$$
$$=\frac{(2a'-1)P(a^{\mathrm{c}}(X))}{P(a^{\mathrm{c}}(X))P(a(X)=a'|a^{\mathrm{c}}(X))}$$
$$=\frac{2a'-1}{P(a(X)=a'|a^{\mathrm{c}}(X))}$$

(2) IATT: Likewise, with $P^*(a^c(X)) = P(a^c(X)|a(X) = 1)$, we can rewrite $\gamma$:

$$\gamma(1, a^c(X)) = \frac{P^*(a^c(X))}{P(a^c(X))P(a(X) = 1|a^c(X))}$$

$$= \frac{P(a^c(X)|a(X) = 1)}{P(a^c(X))P(a(X) = 1|a^c(X))}$$

$$= \frac{P(a(X) = 1|a^c(X))P(a^c(X))}{P(a(X) = 1)P(a^c(X))P(a(X) = 1|a^c(X))}$$

$$= \frac{1}{P(a(X) = 1)}$$

$$\gamma(0, a^c(X)) = -\frac{P^*(a^c(X))}{P(a^c(X))P(a(X) = 0|a^c(X))}$$

$$= -\frac{P(a^c(X)|a(X) = 1)}{P(a^c(X))P(a(X) = 0|a^c(X))}$$

$$= -\frac{P(a(X) = 1|a^c(X))P(a^c(X))}{P(a(X) = 1)P(a^c(X))P(a(X) = 0|a^c(X))}$$

$$= -\frac{P(a(X) = 1|a^c(X))}{P(a(X) = 0|a^c(X))P(a(X) = 1)}$$

$$\gamma(a', a^c(X)) = \frac{a'}{P(a(X) = 1)} - \frac{(1 - a')P(a(X) = 1|a^c(X))}{P(a(X) = 0|a^c(X))P(a(X) = 1)}$$

### A.1.3. A GENERAL $P^*$

Rather than setting a specific $P^*(a^c(X))$, we can identify the isolated effect for any target distribution $P^*$ for which we have a corpus $T$. Consider a corpus $T$ where $a^c(X)$ follows target distribution $P^*$, and a corpus $S$ where $a^c(X)$ follows the initial distribution $P$. Let $C$ be a random variable that indicates which corpus a text comes from.

Then we notice that:

- $P^*(a^c(X))$ can be equivalently written as $P(a^c(X)|C = T)$.

- $P(a^c(X))$ can be equivalently written as $P(a^c(X)|C = S)$.

Then we have

$$\gamma(a', a^c(X)) = \frac{P^*(a^c(X))}{P(a^c(X))P(a(X) = a'|a^c(X))}$$

$$= \frac{P(a^c(X)|C = T)}{P(a^c(X)|C = S)P(a(X) = a'|a^c(X), C = S)}$$

$$= \frac{P(C = T|a^c(X))P(a^c(X))P(C = S)}{P(C = T)P(C = S|a^c(X))P(a^c(X))P(a(X) = a'|a^c(X), C = S)}$$

$$= \frac{P(C = S)}{P(C = T)} \times \frac{P(C = T|a^c(X))}{P(C = S|a^c(X))} \times \frac{1}{P(a(X) = a'|a^c(X), C = S)}$$

All quantities are easily estimated: $P(C = S)$, $P(C = T)$ from sample proportions; $P(C = T|a^c(X))$ and $P(C = S|a^c(X))$ from a classifier trained on both corpora that predicts $C$ given $a^c(X)$ as features; and $P(a(X) = a'|a^c(X), C = S)$ from a classifier trained on corpus $S$ that predicts $a(X)$ given $a^c(X)$ as features.

### A.1.4. Unbiasedness of $\widehat{\tau}_{DR}$ Given One Correct Model

*Proof.* $\mathbb{E}_{Y \sim P_y}[\mathbb{E}_{X,X^*}[\widehat{\tau}_{DR}]] = \tau^*$ when either $\widehat{\gamma}(a', a_s^{\mathsf{c}}(X)) = \gamma(a', a^{\mathsf{c}}(X))$ or $\widehat{g}(a', a_s^{\mathsf{c}}(X)) = g(a', a^{\mathsf{c}}(X))$.

Consider data $D = (X_i, Y_i)$, $X_i \sim P$ and $Y_i \sim P_y$; and $X_j \sim P^*$ ($i \in [n], j \in [m]$).

First, we rewrite:

$$
\mathbb{E}_{Y \sim P_y}[\mathbb{E}_{X,X^*}[\widehat{\tau}_{DR}]] = \mathbb{E}_{Y \sim P_y}\Big[E_{X,X^*}\Big[\frac{1}{m}\sum_{j=1}^{m}\big[\widehat{g}(1, a_s^{\mathsf{c}}(X_j^*)) - \widehat{g}(0, a_s^{\mathsf{c}}(X_j^*))\big]
$$
$$
+ \frac{1}{n}\sum_{i=1}^{n}\widehat{\gamma}(a(X_i), a_s^{\mathsf{c}}(X_i))(Y_i - \widehat{g}(a(X_i), a_s^{\mathsf{c}}(X_i)))\Big]\Big]
$$

$$
= \frac{1}{m}\sum_{j=1}^{m}\mathbb{E}_{Y \sim P_y}[\mathbb{E}_{X^*}\left[\widehat{g}(1, a_s^{\mathsf{c}}(X_j^*)) - \widehat{g}(0, a_s^{\mathsf{c}}(X_j^*))\right]]
$$
$$
+ \frac{1}{n}\sum_{i=1}^{n}\mathbb{E}_{Y \sim P_y}[\mathbb{E}_X[\widehat{\gamma}(a(X_i), a_s^{\mathsf{c}}(X_i))(Y_i - \widehat{g}(a(X_i), a_s^{\mathsf{c}}(X_i)))]]
$$

$$
= \frac{1}{m}\sum_{j=1}^{m}\mathbb{E}_{X^*}\left[\widehat{g}(1, a_s^{\mathsf{c}}(X_j^*)) - \widehat{g}(0, a_s^{\mathsf{c}}(X_j^*))\right]
$$
$$
+ \frac{1}{n}\sum_{i=1}^{n}\mathbb{E}_{Y \sim P_y}[\mathbb{E}_X[\mathbb{1}\{a(X_i) = 1\}\widehat{\gamma}(1, a_s^{\mathsf{c}}(X_i))(Y_i - \widehat{g}(1, a_s^{\mathsf{c}}(X_i)))
$$
$$
+ \mathbb{1}\{a(X_i) = 0\}\widehat{\gamma}(0, a_s^{\mathsf{c}}(X_i))(Y_i - \widehat{g}(0, a_s^{\mathsf{c}}(X_i)))]]
$$
$$
= \frac{1}{m}\sum_{j=1}^{m}\mathbb{E}_{X^*}[\widehat{g}(1, a_s^{\mathsf{c}}(X_j^*))] + \frac{1}{n}\sum_{i=1}^{n}\mathbb{E}_{Y \sim P_y}\left[\mathbb{E}_{X^*}\left[\mathbb{1}\{a(X_i) = 1\}\frac{P^*(a^{\mathsf{c}}(X_i))}{\widehat{P}(a_s^{\mathsf{c}}(X_i), a(X_i) = 1)}(Y_i - \widehat{g}(1, a_s^{\mathsf{c}}(X_i)))\right]\right]
$$
$$
- \left(\frac{1}{m}\sum_{j=1}^{m}\mathbb{E}_{X^*}[\widehat{g}(0, a_s^{\mathsf{c}}(X_j^*))]\right.
$$
$$
\left.+ \frac{1}{n}\sum_{i=1}^{n}\mathbb{E}_{Y \sim P_y}\left[\mathbb{E}_{X^*}\left[\mathbb{1}\{a(X_i) = 0\}\frac{P^*(a^{\mathsf{c}}(X_i))}{\widehat{P}(a_s^{\mathsf{c}}(X_i), a(X_i) = 0)}(Y_i - \widehat{g}(0, a_s^{\mathsf{c}}(X_i)))\right]\right]\right)
$$

Case 1: $\widehat{\gamma}(a', a_s^{\mathsf{c}}(X)) = \gamma(a', a^{\mathsf{c}}(X))$ (i.e., $\frac{P^*(a^{\mathsf{c}}(X))}{\widehat{P}(a_s^{\mathsf{c}}(X), a(X) = a')} = \frac{P^*(a^{\mathsf{c}}(X))}{P(a^{\mathsf{c}}(X), a(X) = a')}$).

First, we consider the IPW term:

$$
\frac{1}{n}\sum_{i=1}^{n}\mathbb{E}_{Y \sim P_y}\left[\mathbb{E}_X\left[\mathbb{1}\{a(X_i) = a'\}\frac{P^*(a^{\mathsf{c}}(X_i))}{\widehat{P}(a_s^{\mathsf{c}}(X_i), a(X_i) = a')}(Y_i - \widehat{g}(a', a_s^{\mathsf{c}}(X_i)))\right]\right]
$$
$$
= \frac{1}{n}\sum_{i=1}^{n}\mathbb{E}_{Y \sim P_y}\left[\mathbb{E}_X\left[\mathbb{1}\{a(X_i) = a'\}\frac{P^*(a^{\mathsf{c}}(X_i))}{P(a^{\mathsf{c}}(X_i), a(X_i) = a')}(Y_i - \widehat{g}(a', a_s^{\mathsf{c}}(X_i)))\right]\right]
$$
$$
= \mathbb{E}_{Y(\cdot) \sim \mathcal{G}}\left[\mathbb{E}_X\left[\sum_{x \in \mathcal{X}}\frac{P^*(a^{\mathsf{c}}(x))}{P(a^{\mathsf{c}}(x), a(x) = a')}(Y(a', a^{\mathsf{c}}(x))\right.\right.
$$
$$
\left.\left.- \widehat{g}(a', a_s^{\mathsf{c}}(x)))\mathbb{1}\{a(x) = a'\}\mathbb{1}\{a^{\mathsf{c}}(X) = a^{\mathsf{c}}(x), a(X) = a(x)\}\right]\right]
$$
$$
= \mathbb{E}_{Y(\cdot) \sim \mathcal{G}}\left[E_{X^*}\left[Y(a', a^{\mathsf{c}}(X^*)) - \widehat{g}(a', a_s^{\mathsf{c}}(X^*))\right]\right] \qquad \text{(following } \tau_{DR} \text{ identification)}
$$
$$
= \mathbb{E}_{Y(\cdot) \sim \mathcal{G}}\left[\mathbb{E}_{X^*}\left[Y(a', a^{\mathsf{c}}(X^*))\right]\right] - \mathbb{E}_{Y(\cdot) \sim \mathcal{G}}\left[\mathbb{E}_{X^*}\left[\widehat{g}(a', a_s^{\mathsf{c}}(X^*))\right]\right]
$$
$$
= \mathbb{E}_{Y(\cdot) \sim \mathcal{G}}\left[\mathbb{E}_{X^*}\left[Y(a', a^{\mathsf{c}}(X^*))\right]\right] - \mathbb{E}_{X^*}\left[\widehat{g}(a', a_s^{\mathsf{c}}(X^*))\right]
$$

Next, we can rewrite the outcome modeling term:

$$\frac{1}{m}\sum_{j=1}^{m}\mathbb{E}_{X^*}[\widehat{g}(a', a_s^{\mathsf{c}}(X_j^*))] = \mathbb{E}_{X^*}\left[\sum_{x\in\mathcal{X}}\widehat{g}(a', a_s^{\mathsf{c}}(x))\mathbb{1}\{a^{\mathsf{c}}(X^*) = a^{\mathsf{c}}(x)\}\right]$$

$$= \sum_{x\in\mathcal{X}}\widehat{g}(a', a_s^{\mathsf{c}}(x))\mathbb{E}_{X^*}\left[\mathbb{1}\{a^{\mathsf{c}}(X^*) = a^{\mathsf{c}}(x)\}\right]$$

$$= \sum_{x\in\mathcal{X}}\widehat{g}(a', a_s^{\mathsf{c}}(x))P^*(a^{\mathsf{c}}(x))$$

$$= E_{X^*}\left[\widehat{g}(a', a_s^{\mathsf{c}}(X^*))\right]$$

So now we have

$$\mathbb{E}_{Y\sim P_y}[\mathbb{E}_{X,X^*}[\widehat{\tau}_{DR}]] = E_{X^*}\left[\widehat{g}(1, a_s^{\mathsf{c}}(X^*))\right] + \mathbb{E}_{Y(\cdot)\sim\mathcal{G}}\left[E_{X^*}\left[Y(1, a^{\mathsf{c}}(X^*))\right]\right] - \mathbb{E}_{X^*}\left[\widehat{g}(1, a_s^{\mathsf{c}}(X^*))\right]$$

$$- \left(\mathbb{E}_{X^*}\left[\widehat{g}(0, a_s^{\mathsf{c}}(X^*))\right] + \mathbb{E}_{Y(\cdot)\sim\mathcal{G}}\left[E_{X^*}\left[Y(0, a^{\mathsf{c}}(X^*))\right]\right] - \mathbb{E}_{X^*}\left[\widehat{g}(0, a_s^{\mathsf{c}}(X^*))\right]\right)$$

$$= \mathbb{E}_{Y(\cdot)\sim\mathcal{G}}\left[E_{X^*}\left[Y(1, a^{\mathsf{c}}(X^*))\right]\right] - \mathbb{E}_{Y(\cdot)\sim\mathcal{G}}\left[E_{X^*}\left[Y(0, a^{\mathsf{c}}(X^*))\right]\right]$$

$$= \mathbb{E}_{Y(\cdot)\sim\mathcal{G}}\left[\mathbb{E}_{a^{\mathsf{c}}(X^*)\sim P^*}\left[Y(1, a^{\mathsf{c}}(X^*)) - Y(0, a^{\mathsf{c}}(X^*))\right]\right]$$

$$= \tau^*$$

Case 2: $\widehat{g}(a', a_s^{\mathsf{c}}(X)) = g(a', a^{\mathsf{c}}(X))$.

Again, we consider the IPW term:

$$\frac{1}{n}\sum_{i=1}^{n}\mathbb{E}_{Y\sim P_y}\left[\mathbb{E}_X\left[\mathbb{1}\{a(X_i) = a'\}\frac{P^*(a^{\mathsf{c}}(X_i))}{\widehat{P}(a_s^{\mathsf{c}}(X_i), a(X_i) = a')}(Y_i - \widehat{g}(a', a_s^{\mathsf{c}}(X_i)))\right]\right]$$

$$= \frac{1}{n}\sum_{i=1}^{n}\mathbb{E}_{Y\sim P_y}\left[\mathbb{E}_X\left[\mathbb{1}\{a(X_i) = a'\}\frac{P^*(a^{\mathsf{c}}(X_i))}{\widehat{P}(a_s^{\mathsf{c}}(X_i), a(X_i) = a')}(Y_i - g(a', a^{\mathsf{c}}(X_i)))\right]\right]$$

$$= \mathbb{E}_{Y(\cdot)\sim\mathcal{G}}\left[\mathbb{E}_X\left[\sum_{x\in\mathcal{X}}\mathbb{1}\{a(x) = a'\}\frac{P^*(a^{\mathsf{c}}(x))}{\widehat{P}(a_s^{\mathsf{c}}(x), a(x) = a')}(Y(a', a^{\mathsf{c}}(x))\right.\right.$$

$$\left.\left. - g(a', a^{\mathsf{c}}(x)))\mathbb{1}\{a^{\mathsf{c}}(X) = a^{\mathsf{c}}(x), a(X) = a(x)\}\right]\right]$$

$$= \mathbb{E}_{Y(\cdot)\sim\mathcal{G}}\left[\sum_{x\in\mathcal{X}}\frac{P^*(a^{\mathsf{c}}(x))}{\widehat{P}(a_s^{\mathsf{c}}(x), a(x) = a')}(Y(a', a^{\mathsf{c}}(x))\right.$$

$$\left. - g(a', a^{\mathsf{c}}(x)))\mathbb{E}_X\left[\mathbb{1}\{a(x) = a'\}\mathbb{1}\{a^{\mathsf{c}}(X) = a^{\mathsf{c}}(x), a(X) = a(x)\}\right]\right]$$

$$= \mathbb{E}_{Y(\cdot)\sim\mathcal{G}}\left[\sum_{x\in\mathcal{X}}P(a^{\mathsf{c}}(x), a(x) = a')\frac{P^*(a^{\mathsf{c}}(x))}{\widehat{P}(a_s^{\mathsf{c}}(x), a(x) = a')}(Y(a', a^{\mathsf{c}}(x)) - g(a', a^{\mathsf{c}}(X_i)))\right]$$

$$= \sum_{x\in\mathcal{X}}P(a^{\mathsf{c}}(x), a(x) = a')\frac{P^*(a^{\mathsf{c}}(x))}{\widehat{P}(a_s^{\mathsf{c}}(x), a(x) = a')}\mathbb{E}_{Y(\cdot)\sim\mathcal{G}}\left[Y(a', a^{\mathsf{c}}(x)) - g(a', a^{\mathsf{c}}(X_i))\right]$$

$$= \sum_{x\in\mathcal{X}}P(a^{\mathsf{c}}(x), a(x) = a')\frac{P^*(a^{\mathsf{c}}(x))}{\widehat{P}(a_s^{\mathsf{c}}(x), a(x) = a')}\mathbb{E}_{Y(\cdot)\sim\mathcal{G}}\left[Y(a', a^{\mathsf{c}}(x)) - \mathbb{E}_{Y(\cdot)\sim\mathcal{G}}[Y(a', a^{\mathsf{c}}(x))]\right]$$

$$= \sum_{x\in\mathcal{X}}P(a^{\mathsf{c}}(x), a(x) = a')\frac{P^*(a^{\mathsf{c}}(x))}{\widehat{P}(a_s^{\mathsf{c}}(x), a(x) = a')} \cdot 0$$

$$= 0$$

Now looking at the outcome modeling term,

$$
\frac{1}{m} \sum_{j=1}^{m} \mathbb{E}_{X^*}[\widehat{g}(a', a_s^{\mathsf{c}}(X_j^*))] = \frac{1}{m} \sum_{j=1}^{m} \mathbb{E}_{X^*}[g(a', a^{\mathsf{c}}(X_j^*))]
$$

$$
= \mathbb{E}_{X^*}\left[ \sum_{x \in \mathcal{X}} g(a', a^{\mathsf{c}}(x)) \mathbb{1}\{a^{\mathsf{c}}(X^*) = a^{\mathsf{c}}(x)\} \right]
$$

$$
= \sum_{x \in \mathcal{X}} g(a', a^{\mathsf{c}}(x)) \mathbb{E}_{X^*}\left[ \mathbb{1}\{a^{\mathsf{c}}(X^*) = a^{\mathsf{c}}(x)\} \right]
$$

$$
= \sum_{x \in \mathcal{X}} P^*(a^{\mathsf{c}}(x)) g(a', a^{\mathsf{c}}(x))
$$

$$
= \mathbb{E}_{a^{\mathsf{c}}(X^*) \sim P^*}[g(a', a^{\mathsf{c}}(X^*))]
$$

$$
= \mathbb{E}_{a^{\mathsf{c}}(X^*) \sim P^*}[\mathbb{E}_{Y(\cdot) \sim \mathcal{G}}[Y(a', a^{\mathsf{c}}(X^*))]]
$$

So now we have

$$
\mathbb{E}_{Y \sim P_y}[\mathbb{E}_{X,X^*}[\widehat{\tau}_{DR}]] = \mathbb{E}_{Y(\cdot) \sim \mathcal{G}}[\mathbb{E}_{a^{\mathsf{c}}(X^*) \sim P^*}[Y(1, a^{\mathsf{c}}(X^*))]] + 0 - (\mathbb{E}_{Y(\cdot) \sim \mathcal{G}}[\mathbb{E}_{a^{\mathsf{c}}(X^*) \sim P^*}[Y(0, a^{\mathsf{c}}(X^*))]] + 0)
$$

$$
= \mathbb{E}_{Y(\cdot) \sim \mathcal{G}}[\mathbb{E}_{a^{\mathsf{c}}(X^*) \sim P^*}[Y(1, a^{\mathsf{c}}(X^*)) - Y(0, a^{\mathsf{c}}(X^*))]]
$$

$$
= \tau^*
$$

$\square$

### A.1.5. CONFIDENCE INTERVALS FOR $\widehat{\tau}_{DR}$

Consider data $(\widetilde{X}_1, \ldots, \widetilde{X}_k) = (X_1, \ldots, X_n, X_1^*, \ldots, X_m^*)$, $(\widetilde{Y}_1, \ldots, \widetilde{Y}_k) = (Y_1, \ldots, Y_n, 0, \ldots, 0)$. Then following standard procedures for doubly robust estimators (Kennedy, 2024), the estimator for the closed-form variance of $\widehat{\tau}_{DR}$ is derived from the influence function as follows.

$$
\widehat{\mathrm{Var}}(\widehat{\tau}_{DR}) = \frac{1}{n+m} \sum_{k=1}^{n+m} \left( \mathbb{1}\{k > n\}(\widehat{g}(1, a_s^{\mathsf{c}}(\widetilde{X}_k)) - \widehat{g}(0, a_s^{\mathsf{c}}(\widetilde{X}_k)))\frac{n+m}{m} \right.
$$

$$
\left. + \mathbb{1}\{k \leq n\}\widehat{\gamma}(a(\widetilde{X}_k), a^{\mathsf{c}}(\widetilde{X}_k))(\widetilde{Y}_k - \widehat{g}(a(\widetilde{X}_k), a^{\mathsf{c}}(\widetilde{X}_k)))\frac{n+m}{n} - \widehat{\tau}_{DR} \right)^2
$$

For the IATE case, we set $P^*(a^{\mathsf{c}}(X)) = P(a^{\mathsf{c}}(X))$, meaning there is no external $X^*$. Instead, the outcome modeling term is also computed over $i \in [n]$, giving the variance:

$$
\widehat{\mathrm{Var}}(\widehat{\tau}_{DR}) = \frac{1}{n} \sum_{i=1}^{n} \left( \widehat{g}(1, a_s^{\mathsf{c}}(X_i)) - \widehat{g}(0, a_s^{\mathsf{c}}(X_i)) + \frac{2a(X_i) - 1}{P(a(X_i)|a^{\mathsf{c}}(X_i))}(Y_i - \widehat{g}(a(X_i), a^{\mathsf{c}}(X_i))) - \widehat{\tau}_{DR} \right)^2
$$

For the IATT case, we set $P^*(a^{\mathsf{c}}(X)) = P(a^{\mathsf{c}}(X)|a(X) = 1)$, so that there is again no external $X^*$. Instead, the outcome modeling term is computed over the subset of $i \in [n]$ where $a(X_i) = 1$:

$$
\widehat{\mathrm{Var}}(\widehat{\tau}_{DR}) = \frac{1}{n} \sum_{i=1}^{n} \left( \frac{\mathbb{1}\{a(X_i) = 1\}}{P(a(X_i) = 1)}(\widehat{g}(1, a_s^{\mathsf{c}}(X_i)) - \widehat{g}(0, a_s^{\mathsf{c}}(X_i))) \right.
$$

$$
\left. + \left( \frac{a(X_i)}{P(a(X_i) = 1)} - \frac{(1 - a(X_i))P(a(X_i) = 1|a^{\mathsf{c}}(X_i))}{P(a(X_i) = 0|a^{\mathsf{c}}(X_i))P(a(X_i) = 1)} \right)(Y_i - \widehat{g}(a(X_i), a^{\mathsf{c}}(X_i))) - \widehat{\tau}_{DR} \right)^2
$$

Asymptotic normality is established using the CLT (Kennedy, 2024):

$$\frac{\widehat{\tau}_{DR} - \tau^*}{\sqrt{\widehat{\text{Var}}(\widehat{\tau}_{DR})}} \to N(0, 1)$$

which gives us the following confidence intervals:

$$\left( \widehat{\tau}_{DR} - z_{\alpha/2}\sqrt{\widehat{\text{Var}}(\widehat{\tau}_{DR})}, \widehat{\tau}_{DR} + z_{\alpha/2}\sqrt{\widehat{\text{Var}}(\widehat{\tau}_{DR})} \right)$$

## A.2. OVB Metrics

Following Chernozhukov et al. (2024), we can define a "short" version of our isolated effect estimand as the difference between $g(1, \cdot) - g(0, \cdot)$ where we use the "short" representation of the non-focal language, $a_s^c(X)$, in place of the true representation $a^c(X^*)$:

$$\tau_s^* = \mathbb{E}_{a_s^c(X^*) \sim P^*} [g(1, a_s^c(X^*)) - g(0, a_s^c(X^*))]$$

Using the proof in Appendix A.1.1, we also have that

$$\tau_s^* = \tau_{DR_s} = \mathbb{E}_{X^* \sim P^*}[g(1, a_s^c(X^*)) - g(0, a_s^c(X^*))] + \mathbb{E}_D[\gamma(a(X), a_s^c(X))(Y - g(a(X), a_s^c(X)))].$$

This allows us to align our estimand with Chernozhukov et al. (2024), where $g_s$ is the short outcome model ($g(a', a_s^c(X))$ in our setting) and $\alpha_s$ are the short Riesz representer weights ($\gamma(a', a_s^c(X))$ in our setting). Then it follows directly that:

$$\begin{aligned} \sigma^2 &= E_P[(Y - g_s)^2] \\ &= E_P[(Y - g(a(X), a_s^c(X)))^2] \end{aligned}$$

$$\begin{aligned} \nu^2 &= E_P[\alpha_s^2] \\ &= E_P[\gamma(a(X), a_s^c(X))^2] \\ &= 2E_{P^*}[\gamma(1, a_s^c(X^*)) - \gamma(0, a_s^c(X^*))] - E_P[\gamma(a(X), a_s^c(X))^2] \end{aligned}$$

## A.3. OVB Estimators

Following the procedure in Chernozhukov et al. (2024), we construct debiased estimators for $\sigma^2$ and $\nu^2$.

$$\widehat{\sigma}^2 = \sum_{i=1}^n (Y_i - \widehat{g}(a(X_i), a_s^c(X_i)))^2$$

$$\widehat{\nu}^2 = \frac{2}{m} \sum_{j=1}^m (\widehat{\gamma}(1, a_s^c(X_j^*)) - \widehat{\gamma}(0, a_s^c(X_j^*))) - \frac{1}{n} \sum_{i=1}^n \widehat{\gamma}(a(X_i), a_s^c(X_i))^2$$

Then the OVB bounds $(\widehat{\tau}_{DR}^-, \widehat{\tau}_{DR}^+)$ on the isolated effect estimate are:

$$\widehat{\tau}_{DR}^-(C_Y, C_D), \widehat{\tau}_{DR}^+(C_Y, C_D) = \widehat{\tau}_{DR} \pm \sqrt{\widehat{\sigma}^2 \widehat{\nu}^2} C_Y C_D$$

# B. Additional Results

## B.1. Additional Interventions (Linear Amazon Outcome)

In this section, we provide and discuss isolated effect estimates for additional interventions on the Amazon dataset (Figure 5). As is the case for the results in the main paper, we see here that as the dimensionality of $a_s^c(X)$ increases, fidelity improves

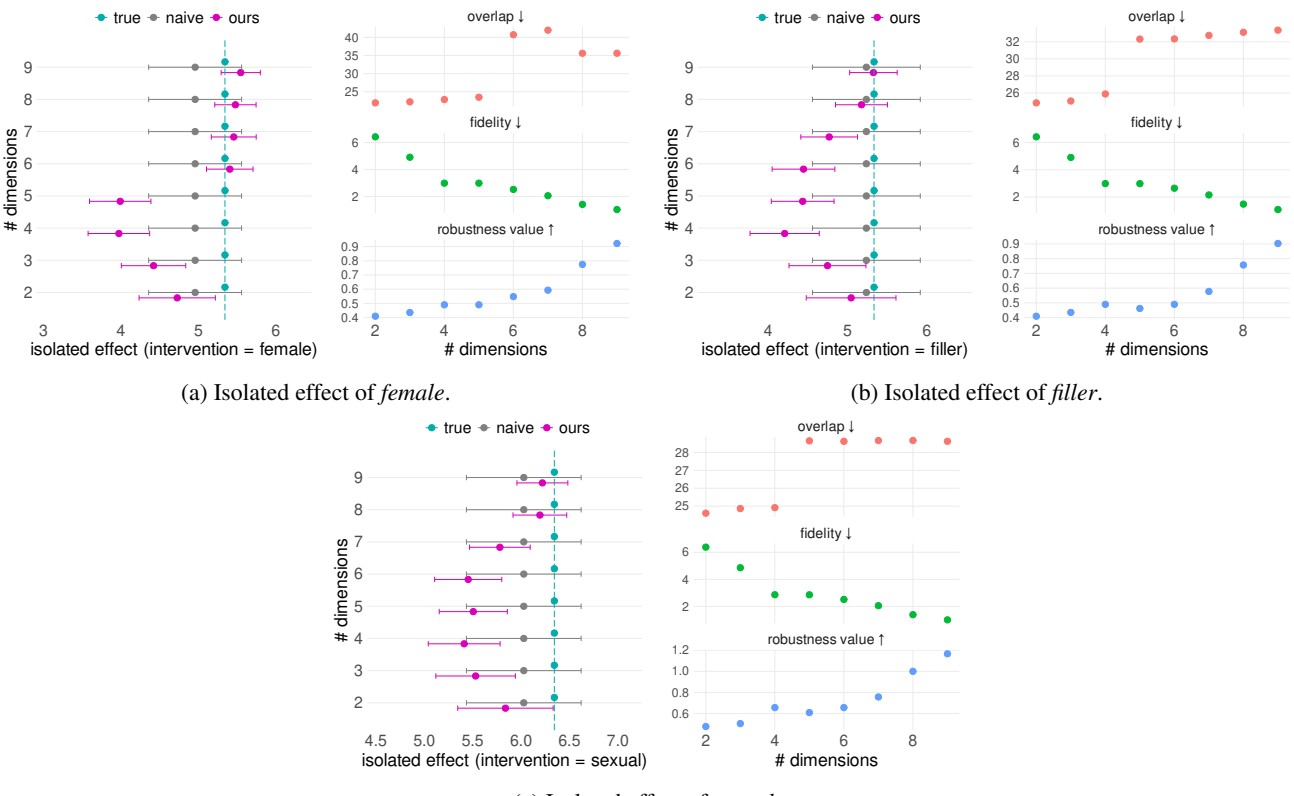

(a) Isolated effect of *female*.

(b) Isolated effect of *filler*.

(c) Isolated effect of *sexual*.

*Figure 5.* Isolated causal effects of linguistic attributes on helpfulness in the Amazon dataset (linear semi-synthetic outcome). Error bars correspond to 95% confidence intervals.

(i.e., the fidelity metric decreases) while overlap becomes worse (i.e., the overlap metric increases).[4] The robustness value suggests overall improvement with increasing dimensionality, though we again note that this may not be the case for some datasets where overlap violations outweigh fidelity improvements.

Interestingly, for all three interventions, we observe that as the number of dimensions increases from 2 to around 5, the isolated effect estimates do not move closer to the ground truth (the point estimates actually move farther, but their confidence intervals suggest that this is not statistically significant). Only after 5 dimensions does the proximity of the estimates to the ground truth increases with dimensionality. We observe this to be the case for the *netspeak* intervention shown in the main paper as well. We speculate that this may be due to the way in which the $n$-dimensional non-focal language representations are constructed. The $n$-dimensional $a_s^c(X)$ representation is always the *same* $n$ lexical categories rather than a random sample of $n$ out of the 9 categories. Therefore, it is possible that the specific additional categories included in the 3- to 5-dimensional representations do not provide much additional information about the outcome, explaining the behavior of the estimates.

### B.2. Nonlinear Amazon Outcome

In this section, we discuss results of isolated effect estimation on a more complex version of the Amazon dataset where the semi-synthetic outcome is a nonlinear function of $a(X), a^c(X)$. The outcome in this setting differs from the one described in Section 4.1 only in that a nonlinear gradient boosting model is used to predict helpful vote count from $a(X), a_s^c(X)$ instead of a linear regression model; this prediction is then noised to obtain the semi-synthetic outcome.

Using this new outcome, we follow the protocol described in Section 5.1 to obtain effect estimates for the two attributes featured in Figure 2: *home* and *netspeak*. During estimation, we use simple feedforward neural networks with no more than

---

[4]The last several dimensions for *female* are an exception to this, where overlap slightly improves—this may be by chance due to better regularization in the classifier models.

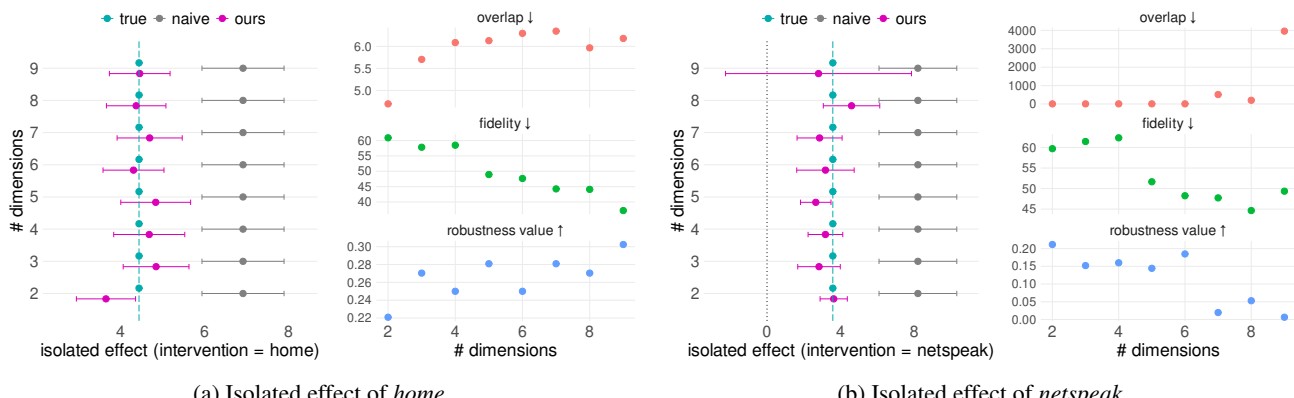

(a) Isolated effect of *home*.  (b) Isolated effect of *netspeak*.

*Figure 6.* Isolated causal effects of linguistic attributes on helpfulness in the Amazon dataset (nonlinear semi-synthetic outcome). Error bars correspond to 95% confidence intervals.

3 layers to fit our importance weight and outcome models.

The results of these additional experiments are consistent with those included in the main paper. For both interventions, we observe that the point estimate of the isolated effect generally grows closer to the ground truth as the number of dimensions increases (i.e., as the number of omitted variables decreases). Likewise, the behavior of the fidelity and overlap metrics $\widehat{\sigma}^2$ and $\widehat{\nu}^2$ remains consistent with expectations: as dimensionality increases, so does $\widehat{\nu}^2$, while $\widehat{\sigma}^2$ decreases. Occasionally slight variability in these trends appears, as we would expect from the noisiness of estimation with more complex models.

Turning to robustness, we see that for *home*, robustness value generally increases with the number of dimensions, suggesting that gains in model fidelity outweigh losses in overlap. For *netspeak*, robustness value remains about the same up until 6 features, then decreases sharply. This coincides with the effect point estimate starting to move away from the ground truth, though the estimate's confidence intervals do still contain the true effect. This suggests that the worsening overlap outweighs gains in model fidelity and that for this effect, the optimal non-focal language representation $a_s^c(X)$ may contain only 6 features.

Finally, we note that once the final feature is added for netspeak, $\widehat{\nu}^2$ sharply increases, signaling much worse overlap. This is an interesting illustration of how soft overlap violations can occur once sufficient information is contained in $a_s^c(X)$ such that the model can fully predict a(X).

# C. Experiments

## C.1. Data

*Table 1.* Composition of data splits and licensing information.

|        | Samples per fold | # folds | License |
|--------|------------------|---------|---------|
| Amazon | 1,000            | 5       | Unknown |
| SvT    | 1,012            | 5       | Unknown |

## C.2. Language Representation Implementation

To implement our lexicons, we use the third-party `liwc` Python library and the `empath` library released by its creators. SenteCon-LIWC and SenteCon-Empath representations are obtained using the `sentecon` library released by its creators. BERT and RoBERTa embeddings are obtained via the HuggingFace `transformers` library using the pre-trained models `bert-base-uncased` and `roberta-base`, respectively. MPNet and MiniLM embeddings are obtained via the HuggingFace `sentence-transformers` library using the pre-trained models `all-mpnet-base-v2` and `all-MiniLM-L6-v2`, respectively. Finally, LLM (GPT-3.5) prompting covariates are taken directly from the SvT dataset

*Table 2.* Technical details for language representation implementations.

|  | Language | Library | Version | Model |
|---|---|---|---|---|
| LIWC | Python | `liwc` | 0.5.0 | - |
| Empath | Python | `empath` | 0.89 | - |
| SenteCon | Python | `sentecon` | 0.1.9 | - |
| BERT embedding | Python | `transformers` | 4.32.1 | `bert-base-uncased` |
| RoBERTa embedding | Python | `transformers` | 4.32.1 | `roberta-base` |
| MiniLM embedding | Python | `sentence-transformers` | 2.2.2 | `all-MiniLM-L6-v2` |
| MPNet embedding | Python | `sentence-transformers` | 2.2.2 | `all-mpnet-base-v2` |
| GPT-3.5 prompting | - | - | - | `gpt-3.5-turbo` |

released by Dhawan et al. (2024). Additional technical details are provided in Table 2.

### C.3. Model Details and Hyperparameters

All outcome models and $a(X)$ classifiers are implemented using the `scikit-learn` Python library (version 1.3.0). Gradient boosting models use a `subsample` proportion of 0.7, i.e., 70% of training samples are used to fit the individual base learners. Neural networks used for outcome models in the nonlinear Amazon setting are implemented with the `MLPRegressor` class and tuned over the following possible layer counts and sizes: (128,), (128, 128), (128, 256, 128).

Logistic and linear regression models are optimized for $L_1$ ratio over the range [0.0, 0.1, 0.5, 0.7, 0.9, 0.95, 0.99, 1.0], where 1.0 corresponds to $L_1$ penalty only and 0.0 corresponds to $L_2$ penalty only. Logistic regression models are further tuned for $C$ (inverse regularization strength) over the following search space: [0.001, 0.01, 0.1, 1.0, 10, 100]. For all interventions, the optimal hyperparameters are a linear regression $L_1$ ratio of 0.5, logistic regression $L_1$ ratio of 0.0, and $C$ of 0.001.

Additionally, the naive estimator is computed formally as follows:

$$\widehat{\tau}_{\text{naive}} = \frac{1}{n} \sum_{i=1}^{n} (a(X_i)Y_i - (1 - a(X_i))Y_i)$$

### C.4. OVB Lower Bound Analysis (Computing $C_Y$ and $C_D$)

Here, we describe our method for computing the explanatory power lost by omitting information from our SenteCon-Empath non-focal language representation.

First, let $a_s^c(X)_{SE}$ denote the "full" SenteCon-Empath representation (i.e., containing all lexical categories and representing the unmasked text). Now let $a_s^c(X)_{SE-}$ denote a SenteCon-Empath representation with omitted information.

Then following Chernozhukov et al. (2024), the explanatory power lost from this omitted information can be computed explicitly as $C_Y$ and $C_D$:

$$C_Y = \sqrt{\frac{\mathbb{E}_D[(g(a(X), a_s^c(X)_{SE}) - g(a(X), a_s^c(X)_{SE-}))^2]}{\mathbb{E}_D[(Y - g(a(X), a_s^c(X)_{SE-}))^2]}}$$

$$C_D = \sqrt{\frac{\mathbb{E}_D[\gamma(a(X), a_s^c(X)_{SE})^2] - \mathbb{E}_D[\gamma(a(X), a_s^c(X)_{SE-})^2]}{\mathbb{E}_D[\gamma(a(X), a_s^c(X)_{SE-})^2]}}$$

We construct representations $a_s^c(X)_{SE-}$ in which each of the labeled lexical categories (`movement`, `science`, `exercise`, and `healing`) is omitted, as well as a representation of the masked text. We then fit outcome models and $a(X)$ classifiers using each $a_s^c(X)_{SE-}$, following the same model fitting methodology described in the main paper, and obtain $g(a(X), a_s^c(X)_{SE-})$ and $\gamma(a(X), a_s^c(X)_{SE-})$. These are used to compute $C_Y$ and $C_D$ over $D$.

## C.5. Computing Resources

All experiments were conducted on consumer-level machines. Experiments involving language models, such as those with MPNet and SenteCon embeddings, were conducted using consumer-level NVIDIA GPUs.

