# OpenReview forum: "Isolated Causal Effects of Natural Language"
_ICML.cc/2025/Conference — ICML 2025 poster_

### Official Review · Reviewer_ZnfT · 2025-02-20

**Overall Recommendation:** 4

**Summary:**

Effects of the attribute of a text on an outcome can be influenced by the surrounding linguistic context around this attribute. This motivates defining an "isolated causal effect of this attribute (the "focal" text), where the surrounding context is marginalized (the "non-focal" text). A framework for estimation of this treatment effect is introduced, with assumptions, identification, a doubly-robust estimator and a sensitivity analysis. Notably, confidence intervals for the estimated isolated effect typically include the ground-truth isolated effect, in stark contrast to the vanilla natural effects from the literature.

(EDIT : updated score based on authors' rebuttal)

**Claims And Evidence:**

I find that while the framework is interesting, there lacks justification for *why* these isolated effects should be considered, in contrast to the natural effect. In the Amazon dataset, the estimator recovers the true isolated effect than the naive method that estimates the natural effect : sure, but the isolated effect was defined as such in the semi-synthetic dataset and the first estimator was inherently designed to estimate it. However, it is interesting to see that the true causal effect on the SvT dataset is the result of an RCT and the estimator developed by authors recovers it unlike the naive method. This suggests that the RCT-given causal effect is the isolated effect and not the natural effect. Thus, if I did not miss anything, I find that authors should expand on this and include more justification (which could be simple citation of previous literature) on why the isolated effect matches such a "ground truth" effect, unlike the natural effect.

Other than that, all other claims seem well justified.

**Essential References Not Discussed:**

I'm not aware of such missing references

**Experimental Designs Or Analyses:**

The analysis in 5.2 seems to match the practice of the Chernozhukov et al. (2024) reference in the paper.

**Methods And Evaluation Criteria:**

Yes, they seem natural.

**Other Comments Or Suggestions:**

- IMO $C_Y$ and $C_D$ should be defined in the main text.

- Authors might include more justification on why these robustness values actually measure... robustness.

**Other Strengths And Weaknesses:**

Strengths :

- The paper is generally clear.
- The perspective is interesting.

Weaknesses :

- There might be a lack of novelty in the treatment effect estimation part, as I do not see how the proposed framework differs from previous work in treatment effect estimation if one posits $T' = a(X)$, $X' = a^c(X)$, $Y'(t) = Y(T'=t, X')$ and performs classical treatment effect estimation, including OVB from Chernozhukov et al. (2024) using $X'$ as covariates, $T'$ as treatment, $Y'(t)$ as potential outcomes. Notably, if I understood correctly, $a(X)$ and $a^c(X)$ are not estimated but given, although $a^c(X)$ is too high-dimensional for direct usage in estimation, thus "covariates" $X'$ and "treatments" $T'$ are given as in any vanilla treatment effect estimation task.

**Questions For Authors:**

I refer to my two main concerns:

(1) Can you include a justification for why the proposed isolated causal effects are more appropriate than natural effects, as outlined in the "Claims and Evidence" section?

(2) Can you explain how your framework differs from vanilla treatment effect estimation, as outlined in the "Other Strengths And Weaknesses" section?

**Relation To Broader Scientific Literature:**

I am not familiar with causal inference for text part, but to the best of my knowledge, all the relevant literature on the treatment effect estimation and OVB part is cited.

One caveat however: the information loss and the confounding error from the two Clivio et al. (2024) references seem to refer to the same quantity.

**Theoretical Claims:**

I checked the proofs and they look correct.

---

> ### Author Rebuttal · Authors · 2025-04-01
>
> Thank you for your positive remarks describing our work as interesting, theoretically correct, and largely well-justified. We address questions and concerns below.
>
> **[Why isolated effects vs. natural effects?]** To illustrate why isolated causal effects are important for language, consider the effect of misinformation in online posts on readers’ voting decisions. Texts containing this type of misinformation also often contain other attributes likely to influence voting—e.g., politically inflammatory messaging. To determine whether action needs to be taken against misinformation, we would need to isolate its effect from the effects of other correlated attributes.
>
> Similarly, in scientific applications, researchers are also frequently concerned with the isolated effect and conduct randomized clinical trials to eliminate effects of potential confounders. In our experiments on the SvT dataset to estimate effects of weight loss medication, we use text data from social media posts, which often encode additional information related to weight loss like users’ exercise habits and diet. We find—as you highlight—that the isolated causal effect estimate from SenteCon-Empath corresponds to the ground truth from real-world clinical trials, while the natural effect estimate does not.
>
> This problem, also known as *aliased treatments*, is an important problem for social scientists studying the effects of language, and prior to our work the proposed solution was to carefully design a text experiment that constructs artificial texts that are not aliased (Fong & Grimmer, 2023). Our work expands the scope of possible research by allowing scientists to use naturally occuring text to study these kinds of effects instead of relying on resource-intensive text experiments with artificial data.
>
> We will include this additional motivation in the introduction and in the discussion of the SvT results.
>
> **[Novelty of proposed framework]** Treatment effect estimation with doubly robust estimators is of course well established in the causal inference literature. A main contribution of our paper is to conceive of, formalize, and parameterize natural language—a complex, unstructured, and high-dimensional form of data—in a way that allows us to use classical causal frameworks like doubly robust effect estimation, which are designed for much simpler forms of data. Indeed, leveraging classical causal estimators to solve new problems in new settings is an active area of research (e.g., Dudik et al. 2011; Schnabel et al. 2016; Azizzadenesheli et al. 2019; Byrd & Lipton 2019; Kallus et al. 2022). Our work introduces the concepts of isolated causal effects and non-focal language, and we define a new problem formulation, estimand, and practical estimation framework for these causal effects. Once language has been formalized in this way, we see the ability to draw on well-studied tools that already exist for estimating robust causal effects as a strength.
>
> Likewise, once language has been parameterized as focal and non-focal components, our OVB estimators indeed naturally follow the work on bounding OVB of Chernozhukov et al. (2024), which presents modern tools for this classical problem. Our main contribution here is to draw the connection between language representation and OVB, which to our knowledge has not previously been articulated. We explore the idea that language representation is lossy specifically in a way that is detrimental to isolated causal effect estimation, and we link that information loss to OVB. We use this connection to introduce OVB-based metrics for evaluating the sensitivity of isolated effect estimates to language representation that are distinct from traditional NLP methods for measuring information loss.
>
> **[Clivio et al. papers]** We will refine the language in our discussion of these references.
>
> **[$C_Y$ and $C_D$]** We originally left the mathematical definitions of $C_Y$ and $C_D$ in Appendix C.4 due to space constraints but will move them to the main paper.
>
> **[Robustness value]** Robustness in causal inference refers to the stability of the effect estimate under potential errors or omissions in modeling, assumptions, and/or data. Our robustness value measures how much OVB can be present before the point estimate of the causal effect changes from the correct to the incorrect sign. A larger robustness value indicates that the effect estimate better tolerates omission of key variables, suggesting that it is stable and therefore robust. We will emphasize this intuition when we define the robustness value in Section 3.3.
>
> We hope that our response addresses any concerns you may have and that you will consider revising your score. Thank you again!
>
> ---
> Azizzadenesheli et al. "Regularized Learning for Domain Adaptation under Label Shifts." ICLR 2019
>
> Dudík et al. "Doubly robust policy evaluation and learning." ICML 2011
>
> Schnabel et al. "Recommendations as treatments: Debiasing learning and evaluation." ICML 2016

---

> > ### Comment · Reviewer_ZnfT · 2025-04-07
> >
> > I can see that I wrote my rebuttal comment as an official comment and not a rebuttal commment.  But it does not change the result:
> >
> > Many thanks, this addresses my concerns. I will move to a clear Accept.

---

> > > ### Author Response · Authors · 2025-04-08
> > >
> > > Thank you again for your feedback and your response to our rebuttal. We appreciate it!

---

### Official Review · Reviewer_sf2t · 2025-03-04

**Overall Recommendation:** 4

**Summary:**

The paper introduces a framework for estimating isolated causal effects of language, which focuses on how specific linguistic attributes influence external outcomes while controlling for non-focal language to mitigate OVB. It uses doubly robust estimators to ensure unbiased estimations. The authors define three metrics (fidelity, overlap, and robustness value) to assess estimation quality. Experiments on a semi-synthetic dataset and a real-world dataset show that SenteCon-based representations yield the most reliable estimates, LLM-Prompting is effective but costly, and MPNet embeddings suffer from overlap violations. The study highlights the fidelity-overlap tradeoff to show the importance of balancing representations for robust causal inference for texts.

**Claims And Evidence:**

I think most claims are well-supported by theoretical and empirical evidence.  This tradeoff concept is valid. But this phenomenon may be general in standard causal inference and classical machine learning where more features can hamper overlap. They do put it into a text-specific lens.

**Essential References Not Discussed:**

No.

**Experimental Designs Or Analyses:**

The experiment design is sound. It is expected that using SenteCon yields the most balanced results as it is a mixture of discrete and continuous embeddings. Adding more experiments using textual embeddings from common LMs might enhance persuasiveness.

**Methods And Evaluation Criteria:**

I think the doubly robust is a good choice for this task. Generally, the methods and evaluation make sense for the problem.

**Other Comments Or Suggestions:**

- Based on the framework of this paper, researchers may expect deeper discussion or more varied experiments of how some representation learning methods can be specifically designed for text tasks to balance the tradeoff (for future work).
- The authors could try some dimension reduction methods for high-dimensional language embeddings, as it is a common approach for handling text embeddings. Reducing dimensionality within a dataset might capture unique non-focal features specific to the data.

**Other Strengths And Weaknesses:**

This is not a big issue, but I have some concerns about the overlap assumption mentioned in the problem setting. For simple tasks, such as binary intervention and low-dimensional focal representation, positivity is easier to achieve. However, this may not hold for more complex tasks, and it might be related to the scale of the data. The subsequent experiments also show that overlap is violated under certain conditions.

**Questions For Authors:**

- What is the exact outcome model?
- Did you try non-synthetic Amazon experiments? Is the original data too random?

**Relation To Broader Scientific Literature:**

This paper builds on some recent research like codebook functions (Egami et al., 2022), natural text experiments (Fong & Grimmer, 2023), and methods that integrate textual features with causal inference (Pryzant et al., 2021). Unlike prior studies that estimate the effect of a focal text feature while allowing correlated text attributes to influence outcomes, this work aims to isolate a binary linguistic attribute by controlling for the nonfocal text distribution. It also extends OVB analysis in high-dimensional settings by introducing fidelity and overlap metrics. These contributions enhance the rigor of text-based causal inference by providing new evaluation metrics and sensitivity checks.

**Theoretical Claims:**

I checked equations in 3.1 and 3.2 and briefly looked at Appendix A. I am not aware of any issues.

---

> ### Author Rebuttal · Authors · 2025-04-01
>
> Thank you for your feedback! We appreciate your positive remarks describing our work as theoretically and empirically well-supported, experimentally sound, and enhancing the rigor of text-based causal inference. We address further comments and questions below.
>
> **[Experiments with additional LM embeddings and dimensionality reduction]** We include additional results ([anonymous link](https://naturl.link/extra-embeds)) on the SvT dataset using embeddings from 3 common LMs: BERT, RoBERTa, and MiniLM. We find that the 3 new LM embeddings improve upon the previous MPNet over all metrics but still do not match SenteCon-Empath.
> - The MiniLM effect point estimate is almost as close to the true effect as SenteCon-Empath, but its robustness value remains middling due to poorer overlap. The BERT and RoBERTa point estimates are only slightly better than MPNet, and their robustness values are lower than MiniLM.
> - BERT and RoBERTa have much better overlap than either MPNet or MiniLM, but worse fidelity. Since BERT and RoBERTa are higher-dimensional than MiniLM, this is surprising but may be due to optimization of MiniLM for sentence-level tasks in the sentence-transformers library. Interestingly, this result suggests the fidelity-overlap tradeoff involves more than just the dimensionality of the representation.
>
> We also include the results of dimensionality reduction using SVD on the MPNet and RoBERTa embeddings, which are the two most complex representations.
> - For both representations, dimensionality reduction improves both the robustness value and the point estimate, highlighting the potential of data-based post-processing of representations.
> - In fact, both point estimates move from outside the true effect confidence interval to inside the true effect confidence interval after SVD.
>
> We thank you for suggesting these experiments and look forward to including them in the paper.
>
> **[Overlap assumption]** It is true that overlap violations can be a concern in language settings (D’Amour et al. 2017). As we mention in our paper, high-dimensional representations with good fidelity are prone to overlap violations, and so we rely on our OVB metrics, overlap and robustness value, to warn us when such violations may be occurring. In this paper, we assume that strict overlap holds (i.e., that there is *some* non-zero chance of each $a(X)$ condition given $a^c(X)$). In our experiments, we have one result that suggests a “soft” overlap violation (i.e., the probability of one $a(X)$ condition is very small but *not* non-zero given $a^c(X)$): the MPNet result on SvT, where the overlap metric is fairly large. We also have a negative overlap value for Empath on the same dataset; however, for this metric we use a doubly robust estimator, and doubly robust estimates do not necessarily satisfy criteria like being non-negative in noisy settings. Therefore, this negative overlap value (which is small in magnitude) may not necessarily be due to an overlap violation.
>
> **[Representations for balancing fidelity-overlap tradeoff]** We agree that the fidelity-overlap tradeoff naturally suggests learning representations that optimize the tradeoff. Experiments that achieve this require further theoretical analysis and empirical work and are outside the scope of this paper—this is in fact the basis of ongoing research—but we are happy to expand on this topic when discussing future work in our conclusion.
>
> **[Exact outcome model]** For Amazon, the outcome model is a linear regression model of the semi-synthetic $Y$ fit over the LIWC categories that form $a(X)$, $a^c(X)$. For SvT, the outcome model is a gradient boosting classifier fit over $a(X)$ and each named non-focal language representation.
>
> **[Non-synthetic Amazon experiments]** We did not use the original Amazon data because the true causal effects of the text interventions are unknown, so there is unfortunately no way to validate effect estimates on this data. The SvT dataset allowed us to validate our methods in a natural setting in which the true effect *was* known through an accompanying external clinical trial. Since such trials are highly resource-intensive, for Amazon we rely on the more common practice of obtaining true effects by generating a semi-synthetic dataset (Veitch et al. 2020; Pryzant et al. 2021). If you are interested in experimental results from a more complex semi-synthetic Amazon dataset, please see our response to reviewer gbt6 titled [Nonlinear $Y$, $a(X)$, $a^c(X)$ relationship].
>
> We hope that our response addresses any concerns you may have and that you will consider revising your score. Thank you again!
>
> ---
> D’Amour et al. "Overlap in observational studies with high-dimensional covariates." J. Econom. 221.2 (2021): 644-654
>
> Pryzant et al. "Causal Effects of Linguistic Properties." NAACL 2021
>
> Veitch et al. "Adapting text embeddings for causal inference." UAI 2020

---

> > ### Comment · Reviewer_sf2t · 2025-04-09
> >
> > Copied from my official comment:
> >
> > Thanks for the authors' responses, which address many of my concerns. I'd like to raise my score.

---

> > > ### Author Response · Authors · 2025-04-09
> > >
> > > Thank you—we appreciate your time and feedback!

---

### Official Review · Reviewer_gbt6 · 2025-03-06

**Overall Recommendation:** 3

**Summary:**

This paper, based on the principle of omitted variable bias, proposes a framework to estimate the sensitivity of bias in evaluating the non-focal language outside of the intervention and the quality of isolated effect estimation along the two key dimensions of  fidelity and overlap.

**Claims And Evidence:**

The writing of this work is unclear and contains a large number of technical terms, such as  language attributes and  focal language, as well as specialized terminology in causal estimation, making it challenging for readers outside the field to clearly understand the paper. Additionally, the motivation of the study is not well-articulated—why is it necessary to study the isolated causal effect? Could the authors further elaborate on this?

**Essential References Not Discussed:**

The reviewer is not familiar with the relevant work in this field and therefore cannot provide further comments.

**Experimental Designs Or Analyses:**

How does the accuracy of isolated effects estimated, using the proposed metrics such as confidence intervals, fidelity, and overlap, perform in more complex situations, such as when Y and $a(X)$, $a^c(X)$ exhibit a nonlinear relationship or when $a(X)$ is not a simple binary variable? Additionally, I noticed that in Figure 3, the confidence intervals of the proposed method on the SvT dataset are quite large, with only the SenteCon-Empath case coming close to the true isolated effect. Does this suggest that the method may not be robust enough across different tasks?

**Methods And Evaluation Criteria:**

1. Could the authors further clarify their methodological contributions? From the paper, both the  doubly robust construction and  effect estimation seem to be a combination of existing techniques (please correct me if I’m wrong). It appears that the authors have merely applied existing methods to a new setting.

2. Could the authors clarify the differences and connections between the method proposed in this paper and approaches that use mediation analysis to analyze specific effects? For example, in [1,2], mediation analysis is used to study the effect of LLMs. Similarly, one could consider treating focal language as the variable of interest and non-focal language as a mediator, as they together constitute the input text, while the goal is to isolate the effect of focal language on the output. This would require a strict disentanglement of the mediating effect introduced by non-focal language. Could the authors further elaborate on how their method differs from the mediation-based approaches mentioned above?

[1] Alessandro Stolfo, Zhijing Jin, Kumar Shridhar, Bernhard Schölkopf, and Mrinmaya Sachan. A causal framework to quantify the robustness of mathematical reasoning with language models. arXiv preprint arXiv:2210.12023, 2022.

[2] Han Y, Xu L, Chen S, et al. Beyond Surface Structure: A Causal Assessment of LLMs' Comprehension Ability[J]. arXiv preprint arXiv:2411.19456, 2024.

**Other Comments Or Suggestions:**

N/A

**Other Strengths And Weaknesses:**

N/A

**Questions For Authors:**

N/A

**Relation To Broader Scientific Literature:**

N/A

**Theoretical Claims:**

This paper does not include additional theoretical explanations for the proposed method. Could the authors further clarify certain formulas, such as the importance weight in Line 144 and the estimand $\tau^*$ in Line 15? How are these formulas derived—are they based on existing methods or newly developed? Since the reviewer is not familiar with this field, the appearance of these formulas feels quite abrupt. If these formulas are based on existing methods, it would be helpful to indicate that.

---

> ### Author Rebuttal · Authors · 2025-04-01
>
> Thank you for your thoughtful questions and feedback! We address your comments and clarify points below.
>
> **[Technical language]** We will revise the paper to more clearly introduce and contextualize technical terms.
>
> **[Motivation for isolated effects]** Due to character limits, please see our response to reviewer ZnfT titled [Why isolated effects vs. natural effects?].
>
> **[Methodological contributions]** Likewise, please see our response to reviewer ZnfT titled [Novelty of proposed framework].
>
> **[Derivation of formulas]** The importance weight in line 144 and estimand $\tau^*$ in line 158 draw from the transportability literature and doubly robust estimation literature, respectively. In Appendix A.1.1, we provide the full derivation of the importance weight and show the equivalence of $\tau^*$ in line 158 to $\tau^*$ in Definition 2.1. We initially left these derivations in the appendix due to space constraints. We agree some technical details may have felt abrupt as a result and will move key derivations to the main paper.
>
> **[Mediation analysis]** We appreciate this insightful question. Though there are parallels in intuition between isolated effects and direct effects in mediation analysis, they are different technical problems. Our setting is *not* a mediation setting:
> - In mediation analysis, such as in the papers you mention, interventions (e.g., math operations) cause mediators (e.g., the text “surface”), and mediators cause outcomes. Only the intervention can be controlled directly in the study, while the mediator cannot.
> - In our setting, the entire text is a high-dimensional treatment where both the focal language $a(X)$ and the non-focal language $a^c(X)$ can be directly intervened on by randomizing the text. $a^c(X)$ is not caused by $a(X)$ and *can* be directly controlled, so it is not a mediator.
>
> **[Nonlinear $Y$, $a(X)$, $a^c(X)$ relationship]** $Y$ and $a(X)$, $a^c(X)$ have a linear relationship only in the Amazon data setting. Our SvT dataset results demonstrate that the true effect can still be recovered even when the relationship between the text and outcome is nonlinear and complex.
>
> To further explore this, we conduct additional experiments ([anonymous link](https://naturl.link/nonlinear) to results) on a new version of the Amazon dataset where $Y$ and $a(X)$, $a^c(X)$ have a nonlinear relationship. We fit a nonlinear gradient boosting model over $a(X)$, $a^c(X)$ on the true helpful vote count, then use the predicted vote count (+noise) as our new semi-synthetic $Y$. Using MLPs with <4 layers to fit importance weights and outcome models, we follow the protocol in Sections 4.2.2 and 5.1 to estimate effects for the attributes featured in Fig. 2.
>
> The results of these additional experiments are consistent with the previous Amazon results:
> - For both *home* and *netspeak*, as # dimensions increases (i.e., as # omitted variables decreases), the effect point estimate grows closer to the ground truth, and overlap becomes worse while fidelity improves. There is slight variability in these trends, as we would expect from the extra noise from more complex models.
> - For *home*, robustness value increases with dimensionality, suggesting that gains in fidelity outweigh losses in overlap. For *netspeak*, robustness value decreases sharply after 6 features, suggesting that worse overlap outweighs improved fidelity. This is especially true on the 9th feature, where the overlap sharply worsens, indicating that $a(X)$ can be almost fully predicted from $a^c(X)$.
>
> We thank you for this question and look forward to including these results in the paper.
>
> **[Complex $a(X)$]** If $a(X)$ is continuous rather than binary, this introduces a new causal problem where it is common to estimate a dose-response curve or incremental effect instead of an average treatment effect. As the study of continuous treatment effects is its own active area of research in causal inference (e.g., Kennedy et al. 2017; Brown et al. 2021), we believe it is beyond the scope of this paper.
>
> **[SvT confidence intervals (CIs)]** Wide CIs suggest that the estimates from the data are noisy (i.e., imprecise) but not necessarily that they are not robust. Robustness in causal inference refers to the stability of the effect estimate under potential errors or omissions in modeling, assumptions, and/or data. Our OVB analysis in Fig. 4 suggests that the SenteCon-Empath estimate (which does have wide CIs) *is* robust in this sense since the point estimate remains correctly positive even when we compromise the estimator by removing important features.
>
> We hope our response addresses any potential concerns and that you will consider revising your score. Thank you again!
>
> ---
> Brown et al. "Propensity score stratification methods for continuous treatments." Stat. Med. 40.5 (2021): 1189-1203
>
> Kennedy et al. "Non-parametric methods for doubly robust estimation of continuous treatment effects." J. R. Stat. Soc. Ser. B Methodol. 79.4 (2017): 1229-1245

---

### Decision · Program_Chairs · 2025-05-01

**Decision:**

Accept (poster)

**Comment:**

This paper introduces a framework for estimating isolated causal effects of language—effects attributable to specific, focal linguistic interventions while controlling for other, non-focal textual content. Grounded in the principle of omitted variable bias (OVB), the paper formalizes the problem setup, applies doubly robust estimation techniques, and proposes diagnostic measures (fidelity, overlap, and robustness value) to assess the sensitivity of effect estimates to imperfect representation of non-focal language. Through experiments on both semi-synthetic Amazon data and a real-world dataset aligned with clinical trial outcomes, the authors demonstrate that their framework can recover isolated effects more faithfully than standard natural effect estimators.

While the technical novelty may not be particularly strong, the framework effectively bridges language modeling, dimensionality reduction, and OVB-based sensitivity analysis, and reviewers appreciated the comprehensive evaluation and strengthened empirical results provided during the rebuttal. Overall, the paper presents a well-motivated and thoughtfully constructed framework that is likely to inspire further work at the intersection of language and causal inference. All reviewers were satisfied with the authors' clarifications and raised their scores.